# Simultaneous Multi-objective Alignment Across Verifiable and Non-verifiable Rewards

**Yiran Shen** [1]   **Yu Xia** [1]   **Jonathan D. Chang** [2]   **Prithviraj Ammanabrolu** [1]

## Abstract

Aligning large language models to human preferences is inherently multidimensional, yet most pipelines collapse heterogeneous signals into a single objective. We seek to answer what it would take to simultaneously align a model across various domains spanning those with: verifiable rewards, non-verifiable subjective preferences, and complex interactive scenarios. Such multi-objective alignment setups are often plagued by individual objectives being at odds with each other, resulting in inefficient training and limited user control during inference. To address these issues, we propose **M**ulti-**A**ction-**H**ead **AL**ignment with PRM-guided Dec**O**ding (**MAHALO**), a unified framework that standardizes PRM training across verifiable and non-verifiable settings for step-level supervision, performs vectorized multi-objective alignment with Multi-Action-Head DPO, and enables controllable inference through objective-specific weighting and PRM-guided decoding. Experiments across math reasoning, human values alignment, and multi-turn tutoring show that MAHALO jointly improves multiple objectives simultaneously with limited interference, while remaining generalizable and adaptable across domains and offering flexible user control at inference time. Our code is available at: https://github.com/pearls-lab/multiobj-align.

## 1. Introduction

The success and widespread deployment of large language models (LLMs) have created opportunities for AI assistance across diverse applications, ranging from mathematical problem solving and question answering to educational tutoring (Brown et al., 2020; Ouyang et al., 2022; Lin et al., 2023). However, these real-world applications often demand that models simultaneously satisfy multiple objectives, which exposes a fundamental challenge that aligning LLMs to human preferences is inherently multidimensional (Askell et al., 2021; Bai et al., 2022; Li et al., 2023). For instance, a question-answering system should provide helpful responses while being harmless (Ganguli et al., 2022; Perez et al., 2022), and an AI education tutor must be able to guide students toward accurate understanding while remaining pedagogically engaging (Maurya et al., 2025; Pal Chowdhury et al., 2024). These scenarios span three distinct categories of alignment targets: domains with verifiable rewards where correctness can be automatically checked (e.g., mathematical accuracy), domains with non-verifiable subjective preferences that require human judgment (e.g., helpfulness), and complex interactive scenarios involving multi-turn dialogues (e.g., AI tutoring engagement) where success depends on the downstream impact of the assistant's responses on subsequent user behavior.

Current alignment methods struggle to capture multi-dimensional human preferences. Methods such as reinforcement learning from human feedback (RLHF) (Christiano et al., 2017; Ouyang et al., 2022) distill human comparisons into scalar reward scores for maximizing expected reward. While direct preference optimization (DPO) (Rafailov et al., 2023) eliminates the reward model, it still optimizes along a single preference axis. Both approaches collapse rich, structured human feedback into one-dimensional training signals, discarding valuable trade-off information and resulting in mismatches between nuanced human preferences and simplified optimization objectives.

Several recent works address multi-objective RLHF alignment through linear scalarization (Li et al., 2020; Hu et al., 2023; Zhou et al., 2024; Wu et al., 2023; Guo et al., 2024) or post-hoc parameter merging of specialized models (Rame et al., 2023; Jang et al., 2023). However, these approaches are computationally expensive and typically require retraining when incorporating additional objectives or altering the balance among existing ones. More computationally lightweight methods like MODPO (Zhou et al., 2024) extend DPO to multiple objectives but they still apply fixed dimensional weights during training time, limiting align-

[1]University of California, San Diego [2]Databricks. Correspondence to: Yiran Shen <jes038@ucsd.edu>.

*Proceedings of the 43rd International Conference on Machine Learning*, Seoul, South Korea. PMLR 306, 2026. Copyright 2026 by the author(s).

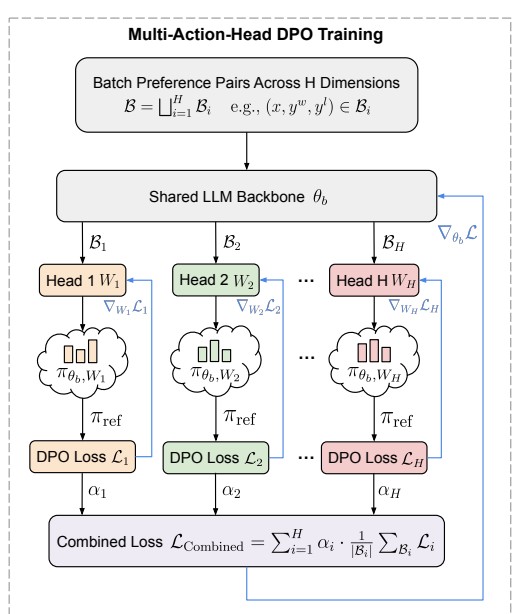 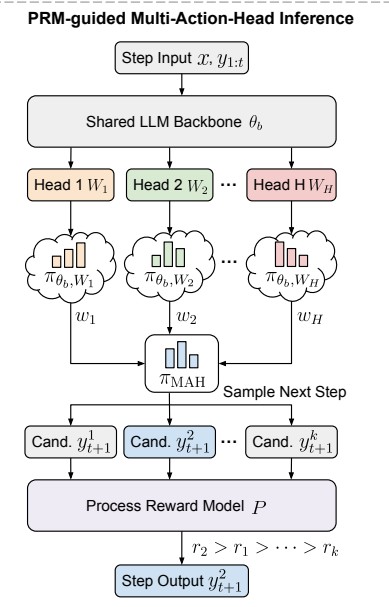

*Figure 1.* Overview of MAHALO, a training-time and test-time framework for multi-objective alignment. Left: we train a single LLM with multiple action heads using head-specific DPO losses and a combined loss on the shared backbone. Right: PRM-guided decoding selects the next step among candidates according to the chosen objective, enabling controllable test-time alignment.

ment flexibility as the dimension weights cannot be changed at inference time. Alternative test-time alignment methods use reward models to guide generation step-by-step but suffer from granularity mismatches between reward definition and generation decisions (Khanov et al., 2024; Deng & Raffel, 2023). For example, outcome reward models are trained to score complete responses while step-level guided decoding operates on partial and incomplete responses, resulting in inconsistencies (Li et al., 2024; Xu et al., 2025). Recent approaches attempt to address this by using more granular reward signals from process reward models (Lightman et al., 2024; Wang et al., 2024b; Luo et al., 2024; Hu et al., 2025; Xiong et al., 2025). However, they mostly focus on verifiable domains where intermediate steps can be reliably evaluated (Zhang et al., 2025b; Zheng et al., 2025) and training PRMs in non-verifiable domains remains a challenge.

To address these limitations, we frame multi-objective alignment as a joint training and inference problem, and propose **M**ulti-**A**ction-**H**ead **AL**ignment with PRM-guided Dec**O**ding (**MAHALO**), as shown in Figure 1. MAHALO combines a standardized PRM training pipeline, Multi-Action-Head DPO, and PRM-guided decoding for controllable multi-objective alignment. Our experiments highlight three practical takeaways. First, the multi-action-head design provides straightforward control with limited interference, where adjusting head weights smoothly shifts behavior toward chosen objectives while keeping other metrics stable. Second, we find that step-level supervision provides useful guidance for non-verifiable dimensions as well. We also show that process supervision can generalize across

tasks as a single PRM trained on a mix of verifiable and non-verifiable data improves all evaluated objectives and demonstrates transferability. Third, we observe that reward verifiability could guide where to spend optimization effort: when rewards are checkable, test-time PRM-guided search delivers the largest gains, whereas noisier subjective rewards benefit more from multi-objective training that shapes shared representations with controllable objective weights. Together, these findings suggest a coherent recipe for building assistants that remain accurate, safe, and engaging across verifiable and non-verifiable settings. The main contributions of this paper are as follows:

- We develop a standardized PRM training pipeline that systematically addresses the challenge of fine-grained supervision across verifiable and non-verifiable domains.

- We propose vectorized multi-objective alignment via Multi-Action-Head DPO, which preserves the multi-dimensional structure of human preferences during training and enables fine-grained preference dimension control during inference.

- Extensive experiments across math reasoning, human value alignment, and multi-turn AI tutoring demonstrate the effectiveness of our training-time and test-time multi-objective alignment framework with possible synergy.

## 2. Related Work

**Process Reward Model.** Process supervision addresses a core limitation of outcome-only evaluation by giving

rewards to intermediate reasoning steps, helping systems avoid trajectories that look correct but contain logical errors. Canonical approaches involve collecting step-level human annotations for math reasoning tasks and training process reward models on these dense supervision signals (Lightman et al., 2024; Xia et al., 2025). Follow-up work scales supervision with automated or weakly supervised labels like per-step Monte Carlo rollouts or self-generated labels (Wang et al., 2024b; Luo et al., 2024). Beyond standard PRMs, recent variants introduce progress or verifier signals that score both partial correctness and future success (Chen et al., 2025b; Setlur et al., 2025). There are also training objectives that regularize PRMs to improve stability (Zhang et al., 2025a). Practical studies discuss data generation, evaluation pitfalls, and how PRMs differ from value functions that predict eventual solvability from partial traces (Zhang et al., 2025b). Process-level search with stepwise scoring has further been shown to beat outcome-level test-time compute baselines in setups including controlled decoding, tree-structured search, and value-guided search (Mudgal et al., 2024; Liu et al., 2024; Yao et al., 2023; Snell et al., 2025; Setlur et al., 2025; Wang et al., 2025).

**Multi-Objective Alignment.** Multi-objective alignment trains or steers language models for multiple, potentially conflicting objectives such as helpfulness, harmlessness, and honesty (Xie et al., 2025). Standard RLHF pipelines fit a scalar reward and fine-tune with PPO, or use scalarized preference optimization (Ouyang et al., 2022; Rafailov et al., 2023; Yuan et al., 2023; Xia et al., 2024; Dong et al., 2023), which collapses trade-offs into one score. Training-time methods relax this restriction with multi-objective RLHF, multi-objective DPO, or parameter mixing to balance different rewards (Zhou et al., 2024; Rame et al., 2023; Wang et al., 2024a; Yang et al., 2024a; Li et al., 2025). Related work also studies interference across skills, domains, or preference data during post-training. Some methods mitigate conflicts by adjusting SFT data composition, scheduling, data usage, or sample selection (Dong et al., 2024a; Wu et al., 2024; Liang et al., 2025), while others precondition policy-optimization gradients to encourage cross-domain compatibility (Liang et al., 2026). Complementing training-time approaches, test-time alignment enables dynamic objective balancing without retraining. These methods modify token probability distributions using reward guidance and perform search under composite objectives, achieving improvements on preference benchmarks while supporting user-specific customization (Khanov et al., 2024; Chen et al., 2025a; Yang et al., 2024b; Lin et al., 2025).

## 3. Background

To understand the challenges in multi-objective alignment, we examine three representative domains. **Mathematics** is a typical verifiable domain, where correctness can be automatically determined using datasets such as MATH (Hendrycks et al., 2021) and OlympiadBench (He et al., 2024). This setting supports automatic reward assignment at both the outcome and process levels. Prior work has shown that process reward models can provide step-level supervision by validating intermediate reasoning steps (Lightman et al., 2024; Wang et al., 2024b). While accuracy is the primary objective, math problem-solving often involves additional dimensions, such as explanation clarity and pedagogical usefulness for users with different expertise levels. **Human Values** differ fundamentally from mathematical correctness because they are not directly verifiable. Dimensions such as helpfulness, harmlessness, and honesty rely on human judgment and are subjective, context-dependent, and sometimes conflicting (Askell et al., 2021; Bai et al., 2022). Recent datasets such as HelpSteer (Wang et al., 2024c) and UltraFeedback (Cui et al., 2024) provide multidimensional annotations and comparisons across criteria including helpfulness, coherence, and truthfulness. The main challenge in this domain is the lack of automatic verification, which limits the availability of fine-grained supervision and complicates optimization across multiple preference dimensions. **Interactive AI Tutoring** introduces additional complexity by combining objective and subjective goals in multi-turn interactions. Beyond correctness, effective tutoring requires engagement, pedagogical structure, and appropriate scaffolding strategies. Relevant datasets include educational dialogue corpora (Stasaski et al., 2020; Macina et al., 2023) and Socratic questioning resources (Shridhar et al., 2022; Ang et al., 2023). In this setting, response quality is best assessed by its effect on subsequent student behavior and learning progress rather than in isolation. We include an example tutoring dialogue in Appendix G.

## 4. Process Reward Model Training

With varying degrees of verifiability and supervision granularity, we first develop a standardized PRM training framework across domains to support multi-objective alignment.

### 4.1. Verifiable Domains

For tasks with objective correctness criteria, such as math, we augment step-level supervision with outcome signals with a value target estimator to train PRMs that both validate the current intermediate step and predict future correctness.

**Step-level Reward.** Given $y_{1:N} = (y_1, y_2, \ldots, y_N)$, the step-level reward is defined as a correctness signal that captures both textual validity and local logical coherence at step $y_t$ (Lightman et al., 2024; Luo et al., 2024). Common practice for obtaining process reward labels involves a multi-stage sampling and annotation process. For example, in Math-Shepherd (Wang et al., 2024b), multiple comple-

tions are sampled from each intermediate step to the final answer. A step is labeled as correct if at least one completion leads to a correct final solution, and incorrect if all completions result in wrong answers.

**Value Reward with Hindsight Relabeling.** Motivated by experience replay in reinforcement learning (Andrychowicz et al., 2018; Harutyunyan et al., 2019), we perform hindsight relabeling in addition to step-level reward. From each step $y_t$, we roll out to its completion $y_{t+1:} = (y_{t+1}, \dots, y_n)$ and evaluate the final solution to obtain a binary terminal correctness reward $z \in \{0, 1\}$. Then, for step $y_t$, we obtain a step-level reward $r_t$ (from annotations or an existing PRM) and add the discounted terminal correctness reward to form a relabeled reward $\tilde{r}_t$ that attributes credit to the current step. For each $y_t$, we sample $M$ independent rollouts and average their relabeled rewards to produce the value target $V_t^{\text{target}}$, which is used to train the PRM by minimizing the mean squared error of its prediction $p_t$.

$$\tilde{r}_t = r_t + \gamma^{n-t} z, \qquad V_t^{\text{target}} = \frac{1}{M} \sum_{m=1}^{M} \tilde{r}_t^{(m)},$$
$$\mathcal{L}_{\text{PRM}} = \mathbb{E}_{t, y_{1:t}} \left[ \left( p_t - V_t^{\text{target}} \right)^2 \right], \tag{1}$$

where $\gamma \in (0, 1)$ is a discount factor that assigns credits based on temporal distance. The relabeled reward enables the PRM to predict both local step reasoning quality and future solution correctness.

### 4.2. Non-verifiable Domains

For domains lacking objective correctness measures, we adapt our PRM training framework based on the availability of clear process structure and rollout difficulty.

**Case A: Clear Process Structure with Efficient Rollout.** When the task has clearly defined intermediate steps that can be meaningfully evaluated, e.g. engagement in math reasoning process, we employ a rollout-based labeling strategy similar to verifiable domain. We first calibrate an LLM-as-Judge $J$ using a few human-annotated ratings $\hat{R}$ to approximate the expected human judgment, $J(y_{1:t}) \approx \mathbb{E}[\hat{R}]$. Then we sample $M$ completions from each step $y_t$ and evaluate the resulting full trajectories using our calibrated LLM-as-Judge $J$. We label $y_t$ as positive if a majority of its $M$ rollouts are judged positive by $J$, capturing subjective step quality through its tendency to yield acceptable outcomes rather than definite correctness:

$$r_t = \mathbb{I}\left[ \frac{1}{M} \sum_{m=1}^{M} \mathbb{I}\left( J\left( y_{1:t}, y_{t+1:n}^{(m)} \right) = \texttt{pos} \right) > \frac{1}{2} \right]. \tag{2}$$

**Case B: Clear Process Structure with Costly Rollout.** When generating rollouts is costly or difficult, for example multi-turn dialogue which requires real user interactions,

we directly query the LLM-as-Judge $J$ on observed trajectory prefixes to obtain the training label: $r_t = J(y_{1:t})$. This approach trades the robustness of rollout-based evaluation for computational efficiency. One can mitigate the increased label noise inherent in this approach through careful judge calibration, ensemble methods, and multi-annotator agreement when feasible.

**Case C: Unclear Process Structure.** For domains where stepwise decomposition lacks clear structure, such as general question answering tasks, we approximate the process modeling by directly evaluating the partial response with a reward model trained with complete responses. For example, one may collect or reuse available pairwise preference data $\{(y^w, y^l)\}$ to train a Bradley-Terry model (Bradley & Terry, 1952) to score process generation $R_\phi(y_{1:t}) \to \mathbb{R}$. The trained reward model provides a holistic quality assessment that serves as guidance during decoding, approximating the intermediate process supervision even when the process structure is not well defined.

## 5. Training and Test-Time Alignment

To align LLMs for multiple objectives across domains, we propose Multi-Action-Head DPO (MAH-DPO) for training-time optimization (Section 5.1) and use our trained PRM directly for test-time alignment through reward-guided decoding with continuing hidden-state (Section 5.2).

### 5.1. Training-Time: Multi-Action-Head DPO

**Direct Preference Optimization.** DPO (Rafailov et al., 2023) optimizes a policy $\pi_\theta$ against a fixed reference policy $\pi_{\text{ref}}$ using preference pairs $\mathcal{D} = \{(x, y^w, y^l)\}$, where $y^w$ is the preferred response to prompt $x$ and $y^l$ is the dispreferred one. The DPO loss is:

$$\mathcal{L}_{\text{DPO}}(\pi_\theta; \pi_{\text{ref}}) = -\mathbb{E}_{(x, y^w, y^l) \sim \mathcal{D}} \left[ \log \sigma \left( \beta \, \Delta(x, y^w, y^l) \right) \right],$$
$$\Delta(x, y^w, y^l) = \log \frac{\pi_\theta(y^w \mid x)}{\pi_{\text{ref}}(y^w \mid x)} - \log \frac{\pi_\theta(y^l \mid x)}{\pi_{\text{ref}}(y^l \mid x)}, \tag{3}$$

where $\sigma(\cdot)$ is the sigmoid function and $\beta > 0$ is a parameter controlling the strength of the preference signal.

**Multi-Action-Head LLM.** To jointly optimize for $H$ distinct objectives while maintaining computational efficiency, we propose the multi-action-head LLM that extends the base LLM with specialized output layers. We maintain a single shared LLM backbone $\theta_b$, while introducing $H$ distinct linear projection heads, one for each alignment objective. This is more efficient than training $H$ separate models, which would require $H$ times the computational resources and fail to leverage cross-objective synergies.

Specifically, let $h_{\theta_b}(x, y_{1:t}) \in \mathbb{R}^d$ denote the $d$-dimensional hidden state produced by the shared LLM backbone $\theta_b$

for input prefix $(x, y_{1:t})$. Each objective $i \in \{1, \ldots, H\}$ has a dedicated projection head parameterized by matrix $W_i \in \mathbb{R}^{d \times |V|}$ to produce objective-specific logits $z_i(x, y_{1:t}) = W_i^\top h_{\theta_b}(x, y_{1:t})$ and token probability distribution $\pi_{\theta_b, W_i}(y_t \mid x, y_{1:t}) = \mathrm{Softmax}(z_i(x, y_{1:t}))$ where $|V|$ is the vocabulary size. The shared LLM backbone captures general language understanding and generation capabilities, while specialized heads encode objective-specific preferences. During inference, our multi-action-head architecture supports flexible objective control by selecting a specific head $i$ for targeted behavior or ensembling logits from multiple heads for balanced performance:

$$\pi_{\mathrm{MAH}}(y_t \mid x, y_{1:t}) = \mathrm{Softmax}\Big(\sum_{i=1}^{H} w_i \, z_i(x, y_{1:t})\Big), \quad (4)$$

where $w_i \geq 0$ are ensemble weights with $\sum_i w_i = 1$. This flexibility allows adaptation to diverse downstream applications and user preferences without separate training for each objective combination.

**Multi-Action-Head DPO Objective.** We first curate $H$ preference datasets $\{\mathcal{D}_i\}_{i=1}^{H}$, where each $\mathcal{D}_i$ contains preference pairs specifically designed for objective $i$ labeled using our trained PRM or from annotated labels. All heads $W_i$ are initialized from the same language modeling head from the supervised fine-tuned (SFT) LLM $\pi_{\theta_b}$ with small random perturbations to encourage specialization. The reference model $\pi_{\mathrm{ref}}$ retains a frozen copy of the base LLM backbone and the unperturbed SFT head which do not share any parameters with the trainable policy model. During training, examples $(x, y^w, y^l) \in \mathcal{D}_i$ are routed to head $i$, and we compute the objective-specific DPO loss:

$$\mathcal{L}_i(\theta_b, W_i) = -\mathbb{E}_{(x, y^w, y^l) \sim \mathcal{D}_i} \Big[ \log \sigma\big(\beta \, \Delta_i(x, y^w, y^l)\big) \Big],$$

$$\Delta_i(x, y^w, y^l) = \log \frac{\pi_{\theta_b, W_i}(y^w \mid x)}{\pi_{\mathrm{ref}}(y^w \mid x)} - \log \frac{\pi_{\theta_b, W_i}(y^l \mid x)}{\pi_{\mathrm{ref}}(y^l \mid x)}.$$
$$(5)$$

Let a mini-batch during training be partitioned as $\mathcal{B} = \bigsqcup_{i=1}^{H} \mathcal{B}_i$ where $\mathcal{B}_i$ gathers the examples $(x, y^w, y^l)$ assigned to head $i$. The combined loss we minimize is

$$\mathcal{L}_{\mathrm{MAH\text{-}DPO}}(\theta_b, \{W_i\}) = \sum_{i=1}^{H} \alpha_i \cdot \frac{1}{|\mathcal{B}_i|} \sum_{\mathcal{B}_i} \mathcal{L}_i(\theta_b, W_i), \quad (6)$$

where $\alpha_i \geq 0$ are objective weights with $\sum_i \alpha_i = 1$.

### 5.2. Test-Time: PRM-Guided Decoding

We also explore the use of our trained PRM during test-time directly via step-level reward-guided decoding. Existing reward-guided decoding or test-time search methods (Khanov et al., 2024; Liao et al., 2025; Park et al., 2025) typically rebuild the prompt at each step by concatenating the newly selected next generation with previous steps.

However, rebuilding and re-encoding the textual prompt each step can change how the prior context is represented within the hidden state, e.g., differences in tokenization around whitespace, shifts in relative positions, and the placement of special tokens. As a result, the next-token distribution after re-encoding can differ from the one obtained by directly continuing from the previous step and such discontinuity can lead to performance degradation. Therefore, to preserve the generation continuity at hidden-state level, we utilize a running past key–value cache during our PRM-guided decoding. The same hidden state is carried forward, so the continuation distribution follows the true incremental decoding rather than a fresh prompt re-encoding approximation. An overview of PRM-guided decoding is provided in Algorithm 1, with additional details in Appendix C.

**Cache Initialization and Candidate Proposal.** Given a prompt $x$, we run a single forward pass with the policy model $\pi_\theta$ to obtain the initial past key–value cache $\mathrm{kv}_0$ and the first next-token distribution. We set the response to $y_{1:0} = \emptyset$ and the generation step index to $t = 0$. This avoids re-encoding $x$ in later steps and provides the reference state from which all continuations proceed. Then, for each step $t$, we propose $K$ candidates from the current running cache $\mathrm{kv}_t$. For each candidate $k$, we clone $\mathrm{kv}_t$ to a local copy and sample the next token from policy model $\pi_\theta$ while carrying that local cache forward. Sampling stops when the boundary detection criterion $\mathcal{Q}$ triggered. This yields a step generation $y_{t+1}^k$ with its end-state cache $\mathrm{kv}_{t+1}^k$.

**Candidate Selection with PRM and Cache Update.** Each sampled candidate is then evaluated by a PRM $P$. Given the current prefix $y_{1:t}$, the score for candidate $k$ is $r_k = P(x, y_{1:t}, y_{t+1}^k)$. We select $k^\star = \arg\max_k r_k$, append the chosen step generation to the response $y_{1:t+1} = y_{1:t} \parallel y_{t+1}^{k^\star}$, and update the current running cache as $\mathrm{kv}_{t+1} = \mathrm{kv}_{t+1}^{k^\star}$. This commit keeps decoding stateful across segments rather than re-encoding the prompt with textual concatenations. We repeat the above candidate proposal starting from $\mathrm{kv}_t$, PRM scoring, and cache update until an end-of-sequence token appears or a budget is reached. With every iteration advancing from the running cache, the generation remains continuous with respect to model's internal hidden state.

## 6. Experiments

In this section, we evaluate the effectiveness of our training-time MAH-DPO and test-time PRM-guided decoding in aligning LLMs along multiple dimensions simultaneously, and explore the potential synergy between them.

**Datasets, Evaluation, and PRM Training.** We evaluate in our method in three domains. *Math:* MATH (Hendrycks et al., 2021) includes 12,500 challenging high-school competition problems requiring multi-step reasoning, enabling

Table 1. Alignment performance of training-time methods across three datasets.

| Method | Acc | Eng |
|---|---|---|
| Base | 0.7107 | 0.5007 |
| SFT | 0.7300 | 0.5920 |
| Single-Head DPO | 0.7253 | 0.7160 |
| MODPO | 0.7280 | 0.7367 |
| DPO Soup | 0.7260 | 0.7353 |
| MAH-DPO Acc Head | **0.7353** | 0.8667 |
| MAH-DPO Eng Head | 0.7267 | **0.8840** |
| MAH-DPO Ensemble | 0.7247 | 0.8733 |

*(a)* Math

| Method | Help | Honest | Truth |
|---|---|---|---|
| Base | 0.5800 | 0.3042 | 0.1888 |
| SFT | 0.5546 | 0.2998 | 0.1992 |
| Single-Head DPO | 0.6043 | 0.3055 | 0.2014 |
| MODPO | 0.6175 | 0.3477 | 0.2325 |
| DPO Soup | 0.6128 | 0.3217 | 0.2153 |
| MAH-DPO Help Head | 0.6309 | 0.3465 | 0.2239 |
| MAH-DPO Honest Head | 0.6257 | 0.3516 | 0.2303 |
| MAH-DPO Truth Head | 0.6257 | 0.3461 | 0.2286 |
| MAH-DPO Ensemble | **0.6389** | **0.3687** | **0.2478** |

*(b)* Human Values

| Method | Acc | Eng |
|---|---|---|
| Base | 0.6560 | 0.3220 |
| SFT | 0.6793 | 0.3473 |
| Single-Head DPO | 0.7040 | 0.4460 |
| MODPO | **0.7047** | 0.3600 |
| MAH-DPO Acc Head | 0.7007 | 0.4447 |
| MAH-DPO Eng Head | 0.6953 | 0.4480 |
| MAH-DPO Ensemble | 0.6893 | **0.4513** |

*(c)* Socratic Mind

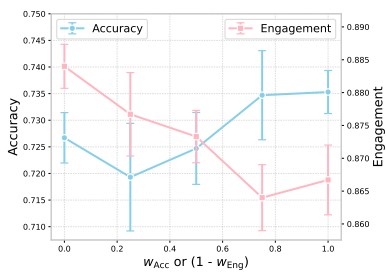

Figure 2. Math results under varying action-head weights.

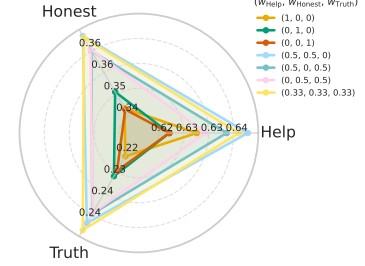

Figure 3. Human Values results under varying action-head weights.

Table 2. Performance on conflicting Human Values subset where helpfulness and honesty preferences disagree.

| Method | Help | Honest |
|---|---|---|
| Base | 0.5800 | 0.3042 |
| SFT | 0.5546 | 0.2998 |
| Single Head DPO | 0.3397 | 0.0734 |
| MODPO | 0.6125 | 0.3386 |
| DPO Soup | 0.6071 | 0.3495 |
| MAH-DPO Help Head | **0.6201** | 0.3503 |
| MAH-DPO Honest Head | 0.6047 | **0.3602** |
| MAH-DPO Ensemble | 0.6123 | 0.3527 |

Table 3. MAH-DPO scaling to more diverse objectives.

| Method | Acc | Eng | Help | Honest | Truth |
|---|---|---|---|---|---|
| Base | 0.7107 | 0.5007 | 0.6466 | 0.4455 | 0.3279 |
| MAH-DPO | **0.7247** | **0.8593** | **0.6528** | **0.4516** | **0.3476** |

verifiable step-level evaluation. *Human Values:* UltraFeedback (Cui et al., 2024) provides preference judgments on helpfulness, honesty, and truthfulness (64k samples). *AI Tutoring Dialogues:* Socratic Mind (Hung et al., 2024) contains multi-turn tutoring conversations for Python programming, averaging 8 turns per session (1362 dialogues). For evaluation, we measure final-answer accuracy in Math and assess engagement using a calibrated LLM-as-Judge with human annotations. In Human Values, we report helpfulness, honesty, and truthfulness as scored by trained reward models. In Tutoring Dialogues, we simulate the student's next turn after the assistant response and score it with a trained PRM to measure both accuracy and engagement. All experimental results are averaged over 3 independent runs and we report standard deviations in Appendix D. We report further evaluation robustness analysis, including cross-model validation and human evaluation, in Appendix F. We train PRMs for each domain using the standardized pipeline in Section 4, with full training details in Appendix A.

### 6.1. Training-Time Alignment

**Baselines and Variants.** We compare against the following baselines and MAH-DPO variants. *Base* is the LLM without post-training. *SFT* performs supervised fine-tuning on preferred responses only. *Single-Head DPO* applies DPO to a single primary objective by pooling all dimension-specific preference data. *MODPO* (Zhou et al., 2024) extends DPO to multiple objectives by combining weighted objectives

during training. *DPO Soup* merges parameters from models trained separately for each objective, following Personalized Soup (Jang et al., 2023). For MAH-DPO, *Individual Head* evaluates each specialized head independently, while *Ensemble* averages logits across heads with equal weights for balanced performance. We further analyze MAH-DPO inference with varying weights in Figure 2 and 3.

**Implementation Details.** We construct paired preference datasets in three domains leveraging either PRM scores or annotations. We then train MAH-DPO on `Qwen2.5-7B-Instruct` for Math and Socratic Mind, and on `meta-llama/Llama-3.1-8B-Instruct` for Human Values. To enable controlled comparisons, we use equal objective weights and balanced sampling so no objective dominates training. We choose domain-specific learning rates, batch sizes, and context windows. Full data construction and hyperparameters are provided in Appendix B.

**Finding 1 - MAH-DPO yields the most balanced multi-objective alignment performance.** Table 1 reports alignment results for all training-time methods. Across tasks, specialized action heads consistently achieve strong performance on targeted metrics, while the equal-weight ensemble head combines these strengths into the best overall alignment profile. This shows that MAH-DPO supports head-level specialization without degrading non-target objectives much, and that simple ensembling at inference time recovers most individual gains without requiring objective-specific model selection. Compared to single-objective baselines

*Table 4.* Alignment performance of test-time methods across three datasets.

| Method | Acc | Eng | Method | Help | Honest | Truth | Method | Acc | Eng |
|---|---|---|---|---|---|---|---|---|---|
| Base | 0.6853 | 0.5133 | Base | 0.5750 | 0.3036 | 0.1904 | Base | 0.6400 | 0.3380 |
| Accuracy PRM-guided | 0.7633 | 0.4720 | Helpful PRM-guided | **0.6706** | 0.4050 | 0.2791 | Accuracy PRM-guided | **0.7127** | 0.2660 |
| Accuracy Value-guided | **0.7993** | 0.4553 | Honesty PRM-guided | 0.6448 | **0.4693** | **0.3383** | Engaging PRM-guided | 0.6507 | **0.4663** |
| Engaging PRM-guided | 0.7013 | **0.7187** | Truthful PRM-guided | 0.6350 | 0.4394 | 0.3296 | | | |
| *(a)* Math | | | *(b)* Human Values | | | | *(c)* Socratic Mind | | |

*Figure 4.* Alignment performance of a unified PRM trained across 7 dimensions in three domains compared with base model and the specialized PRMs trained on each dimension within each domain.

and variants that optimize one dimension at a time, MAH-DPO achieves a more balanced operating point across different objectives, highlighting the benefit of shared representations with lightweight specialization for joint alignment.

**Finding 2 - Head weighting provides smooth control with limited interference.** Figure 2 and 3 show results for varying head weights of MAH-DPO models during inference. Results indicate that adjusting inference-time head weights traces an accuracy–engagement frontier in Math and improves combined outcomes in Human Values. As the engagement weight increases, engagement rises smoothly with only modest accuracy loss; conversely, accuracy-heavy settings retain most of the best accuracy while keeping engagement high. In Human Values, head mixtures attain competitive or best scores across dimensions without sharp degradations on non-emphasized metrics, suggesting that head-level signals in our MAH-DPO trained models interact constructively rather than interfere. In practice, this means we can reliably pick weights to meet application targets without retraining or manual response selection. For example, we can emphasize truthfulness while maintaining helpfulness, or favor engagement while holding accuracy within a narrow band.

**Finding 3 - MAH-DPO remains robust under conflicting objectives and scalable to more diverse objectives.** We conduct further analysis (details in Appendix E) and show in Table 2 and 3 that MAH-DPO maintains strong performance under both explicit objective conflict and increased objective count. Under conflicts between helpfulness and honesty, the specialized heads behave as expected: the Help Head obtains the best helpfulness score and the Honest Head obtains the best honesty score. The fixed equal-weight ensemble no longer dominates both specialized heads, but provides a balanced operating point that remains comparable to or better than scalarization and parameter-merging baselines.

When scaling to a unified five-head setting spanning math and human values objectives, MAH-DPO improves over the base model on all five dimensions simultaneously. This shows that the method remains stable and avoids collapse as objectives grow more numerous and diverse, with a shared backbone learning transferable representations while specialized heads reduce gradient interference.

### 6.2. Test-Time Alignment

**Baselines and Variants.** We report baseline results as well as our PRM-guided decoding variants. *Base* uses the base model directly for stepwise generation without candidate sampling or selection. *Individual PRM-guided Decoding* applies an individual PRM trained for each objective dimension to guide the base model generation step by step following the candidate sampling-then-selection pipeline.

**Implementation Details.** We use the same PRM-guided decoding procedure across domains with the same base models as in training. In Math, we treat \n\n reasoning breaks as step boundaries and guide generation with the accuracy and engagement PRMs. In Human Values, we segment by sentence and paragraph boundaries and score helpfulness, honesty, and truthfulness with trained reward models under fixed per-step and total token budgets. In Socratic Mind, each dialogue turn is a step scored by the engagement and accuracy PRMs.

**Finding 4 - PRM-guided decoding effectively improves the targeted objective.** Table 4 summarizes inference-time PRM-guided decoding results across all datasets. Across settings, PRM guidance consistently increases the selected objective relative to the base model, while leaving non-target metrics close to their original levels rather than causing collapse. This indicates that PRM scores provide localized, well-calibrated signals that influence step-level deci-

*Table 5.* Alignment performance of synergizing training and test-time methods.

| Method | Acc | Eng |
| --- | --- | --- |
| Single-Head DPO | 0.7253 | 0.7160 |
| MODPO | 0.7280 | 0.7367 |
| MAH-DPO | 0.7247 | 0.8733 |
| MAH-DPO + Accuracy Value | **0.8000** | 0.8553 |
| MAH-DPO + Engaging PRM | 0.7207 | **0.9060** |

*(a)* Math

| Method | Help | Honest | Truth |
| --- | --- | --- | --- |
| Single-Head DPO | 0.6043 | 0.3055 | 0.2014 |
| MODPO | 0.6175 | 0.3477 | 0.2325 |
| MAH-DPO | 0.6389 | 0.3687 | 0.2478 |
| MAH-DPO + Help PRM | **0.7165** | 0.4554 | 0.3890 |
| MAH-DPO + Honest PRM | 0.6968 | **0.5196** | **0.4107** |
| MAH-DPO + Truth PRM | 0.6834 | 0.4872 | 0.3630 |

*(b)* Human Values

| Method | Acc | Eng |
| --- | --- | --- |
| Single-Head DPO | 0.7040 | 0.4460 |
| MODPO | 0.7047 | 0.3600 |
| MAH-DPO | 0.6893 | 0.4513 |
| MAH-DPO + Accuracy PRM | **0.7160** | 0.3800 |
| MAH-DPO + Engaging PRM | 0.7120 | **0.5420** |

*(c)* Socratic Mind

sions without introducing strong trade-offs. Compared to unguided decoding, PRM-guided decoding produces controlled and predictable shifts along multi-objective trade-off fronts by selecting among candidate continuations at natural boundaries, improving alignment at test time without requiring any additional training. We provide additional computational analysis, segmentation robustness ablations, and sensitivity studies for continuing-hidden-state decoding in Appendix C.

**Finding 5 - A unified PRM trained on mixed data shows cross-domain effectiveness.** To study whether a single PRM can generalize across domains, we train a unified PRM on a mixture of data covering seven objective dimensions from Math, Human Values, and Socratic Mind (details in Appendix A). Figure 4 shows that this unified PRM improves all objective dimensions relative to the base model across domains. While it does not surpass the best domain-specific PRM on any single axis, it consistently achieves performance between the base model and specialized PRMs, closely tracking the latter in many cases. These results indicate that a generalized PRM can transfer across domains and provide balanced multi-objective improvements without domain-specific retraining or maintaining multiple PRMs.

### 6.3. Synergizing Training and Test-Time Alignment

**Finding 6 - Training and test-time methods are complementary for alignment.** In Table 5 we report results from combining MAH-DPO with ensemble head outputs and PRM-guided decoding at inference time. Across tasks, this pairing consistently moves the joint performance frontier outward relative to training-only baselines, with PRMs enabling targeted improvements along selected dimensions while preserving strong overall performance. The observed trade-offs are smooth and predictable, and in some cases guidance on one dimension also improves correlated objectives, suggesting positive transfer enabled by the disentangled heads learned during training. Overall, MAH-DPO provides a well-structured shared backbone with specialized heads, while PRM guidance steers generation at natural decision points during decoding. Together, they expand the achievable Pareto set and offer practical, fine-grained control at inference time without any additional retraining.

*Table 6.* Effect of continuing hidden state on decoding latency.

| Decoding strategy | w/o cont. hidden state | w/ cont. hidden state | Speedup |
| --- | --- | --- | --- |
| Random Sampling | 47.62s | 9.72s | 4.9x |
| PRM-guided | 165.40s | 39.08s | 4.2x |

*Table 7.* MAH-DPO overhead vs. single-head DPO.

| Configuration | Mean Latency | Memory | Throughput |
| --- | --- | --- | --- |
| Single Head DPO | 9.26s | 14.57 GB | 67.0 tok/s |
| MAH DPO Ensemble (H=2) | 10.46s | 15.61 GB | 66.7 tok/s |
| Overhead | +12.96% | +7.14% | -0.45% |

**Finding 7 - Reward verifiability guides the choice between training and test-time alignment.** Across Tables 1, 4, and 5, a consistent pattern emerges. For objectives with highly verifiable, deterministic rewards, such as Math accuracy, training-time alignment yields only modest gains over strong baselines, while PRM-guided decoding at inference time produces much larger improvements. This indicates that when the signal is precise, step-level scoring at test time can steer generation more effectively than further finetuning. In contrast, for less verifiable or more subjective objectives, including helpfulness, honesty, truth, and engagement, multi-head training already provides substantial benefits by shaping shared representations and separating objectives into specialized heads. Test-time guidance then acts as a refinement mechanism, reweighting or emphasizing specific dimensions without substantially harming others. Overall, highly verifiable rewards benefit most from inference-time search with accurate signals, whereas noisier rewards benefit primarily from multi-objective training, with PRM guidance providing fine-grained control on top.

### 6.4. Computational Overhead Analysis

To assess the efficiency of our training and test-time multi-objective methods, we benchmark latency, memory, and throughput under controlled settings (Qwen2.5-7B-Instruct on 8×H100 GPUs; batch size 1, averaged over 3 runs). As shown in Table 6, the continuing hidden state achieves more than 4× speedups for both random sampling and PRM-guided decoding on 50 MATH samples with 5 candidates per step, confirming that carrying the KV cache eliminates repeated prefix encoding costs (Section 5.2). We also compare MAH-DPO to single-head DPO when using a two-head ensemble. Table 7 shows modest overhead: latency increases by ∼13% and memory by ∼7%, while throughput remains

relatively unchanged. This indicates that MAH-DPO enables multi-objective control with minimal computational cost, without requiring separate models to be trained or stored for each objective.

## 7. Conclusion

In this paper, we present MAHALO, a unified framework for multi-objective alignment during training and at inference time. We standardize process reward model training in both verifiable and non-verifiable settings, propose Multi-Action-Head DPO training with vectorized rewards and pairs the trained model with PRM-guided decoding using a continuing hidden state. Experiments on math reasoning, human value alignment, and multi-turn tutoring domains demonstrate the effectiveness of our framework for multi-objective alignment as well as fine-grained, flexible user control over alignment dimensions. Our framework offers a practical pathway toward AI assistants that are simultaneously accurate, safe, and engaging across diverse domains and applications.

## Acknowledgement

We thank the staff of the School of Computer Science at Georgia Institute of Technology and the members of the Socratic Mind team for their guidance and support with dialogue data collection, dataset interpretation, and appropriate data usage throughout this work.

## Impact Statement

This work supports pluralistic alignment by recognizing that alignment goals vary across people, communities, and contexts, rather than enforcing a single universal target. By unifying learning from objective correctness signals and subjective user preferences, it becomes easier to adapt behavior to different norms and use cases without rebuilding models from scratch. In practice, this can help deployed systems better respect diverse values and communication styles, reduce frustration from misaligned defaults, and give users and deployers clearer control over trade-offs. More broadly, customization and transparent trade-offs can improve accessibility and inclusion, support fairer deployment across groups with different priorities, and encourage more accountable use in socially sensitive settings such as education and public-facing assistance. As with any alignment method, misuse or poorly chosen objective settings can still cause undesirable behavior, so careful evaluation, monitoring, and responsible deployment remain important.

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

# A. PRM Training Details

## A.1. Math PRM Training

**Accuracy PRM Training.** We implement our rollout approach with hindsight relabeling to train a process reward model for mathematical accuracy following Section 4.1. Our method leverages an existing well-trained PRM, specifically `Qwen/Qwen2.5-Math-PRM-7B`, to provide intermediate step-level rewards that we combine with terminal outcome signals through our principled framework. For each candidate reasoning step, we generate 5 independent rollouts using sampling to completion. Step values are computed by combining intermediate PRM rewards with binary final outcome rewards, where correct solutions receive a reward of 1 and incorrect solutions receive 0. These rewards are weighted by a temporal discount factor and averaged across all rollouts to obtain reliable step-level supervision signals for step selection and trajectory extension. The iterative generation process continues until either a final boxed answer is produced or the maximum step limit of 20 is reached, yielding step values within the range $[0, 2]$. Given that the average mathematical problem requires 9-12 reasoning steps, we set the discount rate $\gamma = 0.9$ to appropriately balance immediate step quality assessment with long-term credit assignment.

We also swept the discount factor when turning per-step PRM rewards into value targets and repeated both value-head training and guided decoding. Concretely, for a step prefix $s_{\leq t}$ we formed discounted returns $G_t = \sum_{k \geq 0} \gamma^k r_{t+k}$ with $\gamma \in \{0.9, 0.95\}$, trained the same frozen-backbone + MLP value head to regress $G_t$ via MSE, then used the learned value to steer generation: at each step we propose candidate continuations and pick the one maximizing a blended objective $\alpha V(s_{\leq t} + \text{cand}) + (1 - \alpha) \log P(\text{cand} \mid s_{\leq t})$. Lower $\gamma$ favors short-term gains, while higher $\gamma$ encourages longer-horizon reasoning during decoding.

*Table 8.* Comparison of Math step-level guided decoding methods and their accuracy, averaged over 3 trials.

| Guided Decoding Method | Accuracy | Engagingness |
|---|---|---|
| Baseline step-by-step | 0.6853 ± 0.0163 | 0.5133 ± 0.0543 |
| PRM-guided | 0.7633 ± 0.0050 | 0.7187 ± 0.0266 |
| Value head guided with $\gamma = 0.90$ | 0.7993 ± 0.0172 | 0.4553 ± 0.0221 |
| Value head guided with $\gamma = 0.95$ | 0.7993 ± 0.0081 | 0.5053 ± 0.0050 |
| MAH-DPO Ensemble Head 
 + Accuracy PRM-guided with $\gamma = 0.90$ | 0.8000 ± 0.0231 | 0.8553 ± 0.0136 |
| MAH-DPO Ensemble Head 
 + Accuracy PRM-guided with $\gamma = 0.95$ | 0.7800 ± 0.0197 | 0.8470 ± 0.0098 |

Our PRM architecture follows the design from `Qwen/Qwen2.5-Math-PRM-7B` (Zhang et al., 2025b), where we replace the standard language modeling head with a two-layer scalar value head that produces step-level quality scores. Reasoning steps are serialized using the special separator-token `<extra_0>` in chat-format input, with the transformer's hidden state at each separator token position marking step boundaries. These boundary representations feed into a compact MLP for per-step value prediction. During training, we freeze the PRM backbone parameters from `Qwen/Qwen2.5-Math-PRM-7B` and optimize only the value head using mean squared error loss against the soft step-value targets. Training proceeds for 2 epochs with a batch size of 32 and learning rate of 5e-5.

**Engagement PRM Training.** To evaluate our approach on subjective quality dimensions, we construct an engagement-focused dataset. We sample 50 problems from the MATH training split and generate 4 solution rollouts per problem using the base model. These rollouts use an even mix of engaging and non-engaging reasoning style system prompts to ensure balanced representation (see Appendix H). Human annotators label all 200 responses for engagement quality, providing ground truth supervision for this subjective dimension. We calibrate an LLM-as-Judge using `Qwen/Qwen2.5-72B-Instruct` to evaluate engagement levels, achieving 75.8% classification accuracy against human-labeled solutions. This calibrated judge enables scalable engagement evaluation during PRM training (see Appendix H for the calibrated system prompt).

For each problem, we generate one initial reasoning step, then create eight diverse completions continuing from the current state using generation temperature 1.0. The calibrated LLM-as-Judge scores engagement for every completion batch per step. Following our Case A methodology for non-verifiable domains in Section 4.2, we label a step as engaging if more

than four out of eight rollouts continuing from that step are deemed engaging, otherwise it receives a non-engaging label. This process yields 11.8k step-level engagement annotations. We then convert the training data into incremental reasoning sequences, where each step accumulates the solution path from problem statement through progressive reasoning chains. The base model for the PRM training is `meta-llama/Llama-3.1-8B` configured for binary classification. We train for 2 epochs using a batch size of 128 and a learning rate of 1e-5, which achieves an evaluation accuracy of 92.5%.

### A.2. Human Values PRM Training

Human values represent a non-verifiable domain with no clear process structure. Rather than forcing artificial step-level decomposition, we follow our Case C methodology in Section 4.2 and train a reward model for holistic quality assessment. We train Bradley-Terry reward models on top of the SFT model with base model as `meta-llama/Llama-3.1-8B` following the RLHFlow recipe (Dong et al., 2024b) with learning rate 1e-5 and a batch size of 32 for 3 epochs. The reward model learns to capture human preferences across the helpfulness, honesty, and truthfulness dimensions through pairwise preference optimization, providing dense guidance signals for fine-grained decoding without requiring artificial process supervision.

### A.3. Socratic Mind PRM Training

Students complete post-interaction surveys rating their experience on a 0-6 scale regarding how the Socratic Mind approach enhanced their understanding, serving as our engagement dimension ground truth. We classify ratings $\geq 4$ as engaging interactions. Student dialogues are collected with engagement ratings, and conversations are randomly truncated after assistant turns to create training samples with varying trajectory lengths. We establish calibration datasets with 80 training and 80 test samples to calibrate an LLM-as-judge using GPT-4o (Hurst et al., 2024), achieving 0.8 training accuracy and 0.66 test accuracy for engagement prediction. We additionally curate a specialized judge for accuracy evaluation where system prompt for both objectives can be found in Appendix H. The calibrated LLM-as-judge labels approximately 5k engagement samples and 8k accuracy samples for PRM training, achieving 0.81 test accuracy for engagement and 0.7 test accuracy for accuracy using classification on `Llama-3.1-8B`.

### A.4. Unified PRM Training

We constructed a unified binary-classification corpus by combining all 7 objective dimensions from the domain datasets used in our experiments and formatting each example as a "User:"/"Assistant:" dialogue with blank-line spacing. Math engagement conversations yield incremental stepwise instances labeled as $+/-$. Human value preference pairs are mapped to chosen $= 1$ and rejected $= 0$. Math value scores are normalized per example and thresholded ($> 0.85 \rightarrow 1$, otherwise 0). Socratic Mind engagement and accuracy retain only multi-turn dialogues, with accuracy excluding the last turn. This pipeline produced a total of 168,514 examples with 47.4% positives. We then fine-tuned a pre-trained `Llama-3.1-8B` model with a 2-class classification head using cross-entropy. Training used a batch size of 128, a learning rate of $1 \times 10^{-5}$, and ran for 2 epochs.

# B. Training-time Alignment Details

## B.1. Gradient Analysis

The gradients for the parameters of each head $j$ are isolated by routing, while the backbone LLM gradients accumulate across heads:

$$\nabla_{W_j} \mathcal{L} = \sum_{i=1}^{H} \alpha_i \cdot \frac{1}{|\mathcal{B}_i|} \sum_{(x,y^w,y^l) \in \mathcal{B}_i} \underbrace{\nabla_{W_j} \mathcal{L}_i(\theta_b, W_i; x, y^w, y^l)}_{= 0 \text{ if } j \neq i} = \alpha_j \cdot \mathbb{E}_{\mathcal{B}_j}\left[\nabla_{W_j} \mathcal{L}_j\right],$$

$$\nabla_{\theta_b} \mathcal{L} = \sum_{i=1}^{H} \alpha_i \cdot \frac{1}{|\mathcal{B}_i|} \sum_{(x,y^w,y^l) \in \mathcal{B}_i} \nabla_{\theta_b} \mathcal{L}_i(\theta_b, W_i; x, y^w, y^l). \tag{7}$$

Thus, each head $j$ receives gradients only from its own objective $j$, so token-level conflicts between objectives never directly cancel in the logits for a single head. In contrast, scalarization methods such as MODPO (Zhou et al., 2024) pass all objectives through one policy head, which combines their DPO gradients in the same output layer and can impose a compromise distribution. Although Equation 7 still aggregates gradients via a weighted average, this averaging takes place at the representation level, where objectives can reinforce common structure or learn features that support distinct head behavior, rather than forcing agreement on every token probability.

To achieve more stable training and balanced gradient propagation, we can construct mini-batches with a similar number of examples $|\mathcal{B}_i|$ from each objective $i$ or by tuning the weights $\alpha_i$ when the dataset sizes differ. Since every head consumes the same hidden states for its logits, the computation requires only one backbone forward per input and parallel per-head projections, leveraging cross-objective synergies without introducing excessive extra training cost.

## B.2. Datasets

This work uses three data sources. All components involving human participants or human annotations were reviewed and approved by UC San Diego's Institutional Review Board (IRB) where applicable, and participants provided explicit written consent; participation was voluntary, had no academic consequences, and could be withdrawn at any time. Socratic Mind tutoring dialogues were de-identified, stored with encryption, and accessed only by approved researchers. The public MATH and UltraFeedback datasets were used under their respective licenses, and we cite the original sources. Human annotations were used only to calibrate or validate evaluation models, including Math engagement calibration and human agreement analysis, and Socratic Mind survey-based engagement supervision. We applied content filters and safety checks to reduce risks, avoided generating or encouraging sensitive advice, and will release only code and configurations that do not compromise participant privacy, data licensing, or annotation confidentiality.

## B.3. Math Training Details

Mathematical reasoning presents a natural testbed for multi-objective alignment, as effective tutoring requires balancing computational accuracy with pedagogical engagement. We design our experimental setup to capture this fundamental trade-off in educational AI systems.

**Preference Data Construction.** We construct two complementary preference datasets using the MATH training dataset (12k problems) to target distinct but interrelated aspects of mathematical competence:

- *Accuracy-focused pairs*: For each problem, we generate up to 30 response rollouts using Qwen2.5-7B-Instruct, extract boxed numerical answers, and compare against ground truth solutions. We pair the first correct solution with the first incorrect one encountered, creating 5,574 preference pairs that emphasize computational precision and mathematical correctness.

- *Engagement-focused pairs*: Using the same problem set, we generate 10 rollouts per question and employ LLM-as-Judge evaluation (Qwen2.5-72B-Instruct, temperature=0.1) to assess pedagogical quality. We identify responses that provide clear explanations, intuitive reasoning, and educational insights versus those offering terse or mechanical solutions, yielding 7,930 preference pairs that prioritize learning effectiveness over mere correctness.

This dual construction allows us to examine whether MAH-DPO can simultaneously optimize for mathematical rigor and educational value—objectives that often compete in practice.

**Training Configuration.** We establish a consistent training pipeline across all mathematical experiments. Starting from Qwen2.5-7B-Instruct, we first perform supervised fine-tuning (learning rate $5 \times 10^{-6}$, 2 epochs) to adapt the model to mathematical domains. We then initialize MAH-DPO with small random perturbations (scale=0.001) applied to each head to encourage objective-specific specialization while maintaining shared representations. The multi-head training uses learning rate $1 \times 10^{-6}$, batch size 128, and $\beta = 0.1$, with sequences truncated to 512 prompt tokens and extended to 1536 total tokens to accommodate detailed mathematical reasoning over 2 epochs.

### B.4. Human Values Training Details

Human values alignment represents a more abstract but equally critical challenge, where models must navigate competing ethical principles. We focus on three fundamental dimensions that frequently conflict in real-world applications: helpfulness, truthfulness, and honesty.

**Preference Data Construction.** We leverage the UltraFeedback dataset's rich dimensional annotations to create three targeted preference datasets:

- *Helpfulness*: 59.2k preference pairs contrasting responses that provide comprehensive, actionable guidance versus those offering minimal or irrelevant information.

- *Truthfulness*: 50.8k pairs emphasizing factual accuracy and evidence-based reasoning versus responses containing inaccuracies or unsupported claims.

- *Honesty*: 57.3k pairs focusing on transparent acknowledgment of uncertainty and limitations versus responses that overstate confidence or mask knowledge gaps.

For each dimension, we pair responses with the highest and lowest annotated scores while excluding cases with identical ratings, ensuring clear preference signals. We reserve 2k examples per dimension for comprehensive evaluation across all three values simultaneously.

**Training Configuration.** To maintain experimental consistency while adapting to the distinct characteristics of values alignment, we modify our training approach accordingly. We perform supervised fine-tuning on Llama-3.1-8B using UltraFeedback's preferred responses (learning rate $5 \times 10^{-7}$, 1 epoch, batch size 192) to establish a strong foundation for ethical reasoning. MAH-DPO training employs slightly larger perturbations (scale=0.005) to account for the more nuanced nature of value judgments, with learning rate $5 \times 10^{-7}$, batch size 120, and sequences limited to 256 prompt tokens and 768 total tokens to focus on concise value-aligned responses over 1 epoch.

### B.5. Socratic Mind Training Details

Socratic tutoring illustrates the challenge of multi-objective alignment in educational settings, requiring models to maintain factual accuracy while fostering student engagement through strategic questioning and explanation. This domain tests our approach's ability to handle dynamic, context-dependent trade-offs.

**Preference Data Construction.** We simulate realistic tutoring interactions by randomly sampling 1,000 educational dialogues and introducing natural conversation breakpoints. At each dialogue state, we generate 5 potential assistant responses representing different tutoring strategies—from direct instruction to guided discovery. We then employ trained PRMs specialized for accuracy and engagement assessment to evaluate each candidate response. By selecting the highest and lowest scoring responses for each objective, we create 1,000 preference pairs per dimension that capture the nuanced balance between providing correct information and maintaining pedagogical effectiveness in conversational contexts.

**Training Configuration.** Given the complexity of dialogue understanding, we adopt our mathematical domain configuration while extending context capabilities. We fine-tune Qwen2.5-7B-Instruct (learning rate $5 \times 10^{-6}$, 2 epochs) and apply MAH-DPO with perturbation scale 0.001 to preserve dialogue coherence across heads. Training employs learning rate $1 \times 10^{-6}$, batch size 256, and $\beta = 0.1$, with extended context windows (1336 prompt tokens, 1536 total tokens) to accommodate full dialogue history while maintaining computational efficiency over 2 epochs.

These three experimental domains collectively span the spectrum from concrete mathematical reasoning to abstract value judgments to dynamic conversational interaction, providing a comprehensive testbed for evaluating MAH-DPO's multi-objective alignment capabilities across diverse AI applications.

# C. Decoding-time Alignment Details and Robustness Analysis

## C.1. Computational Analysis

Besides keeping the generation continuity at hidden-state level, our cache-carrying PRM-guided decoding also reduces the computational cost compared to re-encode-per-step baselines. Let $|x|$ be the prompt length, $T$ the committed output tokens, $N$ the number of steps, i.e., detected boundaries, $K$ the candidates per step, and $\bar{L}$ the average candidate length such that $T \approx N\bar{L}$. A re-encode-per-step policy costs $\mathcal{O}(K(|x|N + NT))$ while our cache-carrying policy costs $\mathcal{O}(|x| + KN\bar{L}) = \mathcal{O}(|x| + KT)$. Thus the factor $N$ is removed, enabling better test-time scaling by shifting compute from repeated re-encodings to candidate rollout or longer outputs.

## C.2. Implementation Details

We apply the same PRM-guided decoding procedure across domains using the same base models as in training. In *Math*, we treat reasoning boundaries marked by \n\n as step boundaries and guide step-wise generation with the trained accuracy and engagement PRMs. In *Human Values*, where responses lack a fixed process structure, we impose step boundaries at sentence terminators and paragraph breaks, and score helpfulness, honesty, and truthfulness with trained reward models under a 256-token budget per step and a 1,024-token total budget. In *Socratic Mind*, each dialogue turn forms a step and is scored by the trained engagement and accuracy PRMs. Across all domains, we sample $K=5$ candidates at each step and decode with temperature $= 1.0$, top-p $= 1.0$, and top-k $= 50$. We report additional computation, sensitivity analyses, and continuing-hidden-state results in Appendix C. All results are averaged over three independent runs, with standard deviations in Appendix D.

---

**Algorithm 1:** PRM-Guided Decoding with Continuing Hidden State

**Input:** policy $\pi_\theta$; PRM $P$; boundary detection criteria $\mathcal{Q}$; number of candidates $K$; token budget $T_{\max}$; prompt $x$.

**Output:** response $y$.

$\text{kv}_0 \leftarrow \text{Fwd}_{\pi_\theta}(x); \ y_{1:0} \leftarrow \emptyset; \ t \leftarrow 0.$

**while** $|y_{1:t}| < T_{\max}$ **and** $\text{EOS} \notin y_{1:t}$ **do**

  **for** $k = 1$ **to** $K$ **do**

    $\widetilde{\text{kv}} \leftarrow \text{kv}_t; \ \tilde{y} \leftarrow \emptyset.$

    **while** $\mathcal{Q}(\tilde{y}) = 0$ **do**

      Sample next token $z \sim \pi_\theta(\cdot \mid \widetilde{\text{kv}})$;

      $\widetilde{\text{kv}} \leftarrow \text{Fwd}_{\pi_\theta}(\widetilde{\text{kv}}, z); \ \tilde{y} \leftarrow \tilde{y} \parallel z.$

    Record end-state cache $\text{kv}_{t+1}^k \leftarrow \widetilde{\text{kv}}$;

    Record candidate next step $y_{t+1}^k \leftarrow \tilde{y}$;

    Score with PRM $r_k \leftarrow P(x, y_{1:t}, y^k)$.

  $k^\star \in \arg\max_k r_k$;

  Update running cache $\text{kv}_{t+1} \leftarrow \text{kv}_{t+1}^{k^\star}$;

  Update response $y_{1:t+1} \leftarrow y_{1:t} \parallel y_{t+1}^{k^\star}$;

  $t \leftarrow t + 1.$

---

## C.3. Robustness to Segmentation Strategy

PRM-guided decoding requires a boundary criterion to decide when to score candidate continuations. To evaluate whether our results depend sensitively on the segmentation heuristic, we ablate the boundary strategy in Math and Human Values. In Math, the default uses natural reasoning boundaries marked by \n\n; we compare this with fixed-interval segmentation every 128 or 64 tokens, which may cut across sentences or equations. In Human Values, the default uses both sentence and paragraph boundaries; we compare it with paragraph-only and sentence-only variants.

*Table 9.* Segmentation robustness for PRM-guided decoding.

| Strategy | Acc. (%) | Steps |
|---|---|---|
| \n\n (default) | **79.9** | 7.2 |
| Every 128 tokens | 77.2 | 4.9 |
| Every 64 tokens | 73.0 | 9.3 |

*(a)* Math

| Strategy | Help | Honest | Truth |
|---|---|---|---|
| Sent. + para. (default) | **0.671** | **0.405** | **0.279** |
| Paragraph only | 0.646 | 0.378 | 0.240 |
| Sentence only | 0.662 | 0.374 | 0.253 |

*(b)* Human Values

Table 9 shows that content-aware default boundaries perform best, but alternative or mismatched boundaries lead to graceful rather than abrupt degradation. In Math, fixed 128-token segmentation remains close to the default while reducing the number of scoring steps, whereas overly frequent 64-token segmentation increases the number of decisions and lowers accuracy. In Human Values, paragraph-only and sentence-only segmentation both remain competitive but underperform the combined sentence-and-paragraph heuristic. These results suggest that PRM-guided decoding benefits from content-aware boundaries but is reasonably robust to segmentation choices.

## C.4. Hidden State versus Text Chunk

In this section, we provide further results validating the effectiveness of continuing hidden states in our PRM-guided decoding for alignment. We present comparisons between our continuing hidden state approach and the standard text-chunk concatenation approach, with results shown in Table 10 and 11. From Table 10, we observe that in Human Values where there is not a clear process structure, stepwise generation using text chunk concatenation leads to performance degradation compared to the one-pass generation. Meanwhile, our continuing hidden state approach achieves comparable performance to one-pass generation when no guidance from PRMs is used, and also consistent improvements over text chunk method when guided by PRMs. This demonstrates that text chunk concatenation which requires iterative re-encoding can break the generation continuity while our hidden state approach preserves such continuity for response generation. In Table 11, there is no major performance difference between the text-chunk method and our hidden state method, which indicates that text chunk methods do not break generation continuity when the process structure is clear and well-defined such as in the Math domain.

*Table 10.* Further results of PRM-guided decoding in Human Values: continuing text chunk vs. continuing hidden state.

| Method | Help | Honest | Truth |
| --- | --- | --- | --- |
| One-pass generation without guided decoding (reference) | $0.5800 \pm 0.0066$ | $0.3042 \pm 0.0066$ | $0.1888 \pm 0.0028$ |
| Step-wise generation without guided decoding (text chunk) | $0.4688 \pm 0.0033$ | $0.1857 \pm 0.0016$ | $0.1182 \pm 0.0031$ |
| Step-wise generation without guided decoding (hidden state) | $0.5750 \pm 0.0107$ | $0.3036 \pm 0.0015$ | $0.1904 \pm 0.0036$ |
| Step-wise generation + Helpful PRM guided (text chunk) | $0.6140 \pm 0.0099$ | $0.3273 \pm 0.0069$ | $0.2099 \pm 0.0060$ |
| Step-wise generation + Helpful PRM guided (hidden state) | $0.6706 \pm 0.0093$ | $0.4050 \pm 0.0035$ | $0.2791 \pm 0.0023$ |
| Step-wise generation + Honest PRM guided (text chunk) | $0.6148 \pm 0.0150$ | $0.3860 \pm 0.0106$ | $0.2544 \pm 0.0062$ |
| Step-wise generation + Honest PRM guided (hidden state) | $0.6448 \pm 0.0050$ | $0.4693 \pm 0.0045$ | $0.3383 \pm 0.0025$ |
| Step-wise generation + Truth PRM guided (text chunk) | $0.5775 \pm 0.0155$ | $0.3165 \pm 0.0028$ | $0.2500 \pm 0.0062$ |
| Step-wise generation + Truth PRM guided (hidden state) | $0.6350 \pm 0.0032$ | $0.4394 \pm 0.0036$ | $0.3296 \pm 0.0056$ |

*Table 11.* Further results of PRM-guided decoding in Math: continuing text chunk vs. continuing hidden state.

| Method | Accuracy | Engagement |
| --- | --- | --- |
| One-pass generation without guided decoding (reference) | $0.7107 \pm 0.0090$ | $0.5007 \pm 0.0289$ |
| Step-wise generation without guided decoding (text chunk) | $0.7040 \pm 0.0092$ | $0.4907 \pm 0.0358$ |
| Step-wise generation without guided decoding (hidden-state) | $0.6853 \pm 0.0163$ | $0.5133 \pm 0.0543$ |
| Step-wise generation + Engaging PRM guided (text-chunk) | $0.7187 \pm 0.0147$ | $0.6353 \pm 0.0099$ |
| Step-wise generation + Engaging PRM guided (hidden-state) | $0.7013 \pm 0.0352$ | $0.7187 \pm 0.0266$ |
| Step-wise generation + Accuracy PRM guided (text-chunk) | $0.7973 \pm 0.0083$ | $0.4807 \pm 0.0205$ |
| Step-wise generation + Accuracy PRM guided (hidden-state) | $0.7993 \pm 0.0172$ | $0.4553 \pm 0.0221$ |

## C.5. Sensitivity to PRM Noise on Math

To quantify how sensitive outcomes are to PRM label noise and to study how strongly our framework depends on PRM quality, we run a controlled noise study on MATH across both accuracy and engagement dimensions. During PRM-guided Best-of-$N$ inference, we inject stochastic noise by ignoring the PRM ranking with probability $p$ and instead selecting a candidate uniformly at random. We evaluate $p \in \{0.1, 0.25, 0.5\}$ for both the verifiable objective Accuracy and the non verifiable objective Engagement. Results for both PRM-guided and value-guided accuracy are reported in Table 12, and engagement results are shown in Table 13.

*Table 12.* Math accuracy performance under PRM noise on MATH-500. Retention rate is computed relative to the corresponding clean PRM or value-guided configuration.

| Method | Noise Level $p$ | Accuracy | $\Delta$ from Clean | Retention Rate |
|---|---|---|---|---|
| Accuracy PRM guided | 0.0 (Clean) | 0.7633 | | 100% |
| Accuracy PRM guided | 0.1 (10% noise) | 0.7500 | $-0.0133$ | 98.3% |
| Accuracy PRM guided | 0.25 (25% noise) | 0.7367 | $-0.0266$ | 96.5% |
| Accuracy PRM guided | 0.5 (50% noise) | 0.7247 | $-0.0386$ | 94.9% |
| Accuracy value guided | 0.0 (Clean) | 0.7993 | | 100% |
| Accuracy value guided | 0.1 (10% noise) | 0.7940 | $-0.0053$ | 99.3% |
| Accuracy value guided | 0.25 (25% noise) | 0.7693 | $-0.0300$ | 96.2% |
| Accuracy value guided | 0.5 (50% noise) | 0.7500 | $-0.0493$ | 93.8% |
| Base, no guidance | Clean | 0.6853 | | |

*Table 13.* Math engagement performance under PRM noise on MATH-500. Retention rate is computed relative to the clean PRM-guided configuration.

| Method | Noise Level $p$ | Engagement | $\Delta$ from Clean | Retention Rate |
|---|---|---|---|---|
| Engaging PRM guided | 0.0 (Clean) | 0.7187 | | 100% |
| Engaging PRM guided | 0.1 (10% noise) | 0.6920 | $-0.0267$ | 96.3% |
| Engaging PRM guided | 0.25 (25% noise) | 0.6593 | $-0.0594$ | 91.7% |
| Engaging PRM guided | 0.5 (50% noise) | 0.5980 | $-0.1207$ | 83.2% |
| Base, no guidance | Clean | 0.5133 | | |

**Key Findings**  Performance decreases smoothly as $p$ increases rather than collapsing. Even at $p = 0.5$, which corresponds to coin-flip reliability, PRM-guided accuracy retains $94.9\%$ of its clean score, value-guided accuracy retains $93.8\%$, and engagement retains $83.2\%$ of its clean score. This pattern indicates that the framework remains functional even when the PRM is substantially unreliable.

The framework is robust across both objective types. For moderate noise levels $p \leq 0.25$, retention rates stay above $90\%$ for the verifiable objective Accuracy and for the non-verifiable objective Engagement. This suggests that the same Best-of-$N$ mechanism that stabilizes performance for verifiable metrics also provides meaningful robustness for subjective engagement-oriented behavior.

Despite the injected noise, the method continues to outperform the no-guidance baseline by a substantial margin. Across all noisy configurations, both accuracy and engagement remain well above the baseline on MATH-500, which achieves $68.5\%$ accuracy and $51.3\%$ engagement. This shows that even partially reliable PRMs are sufficient to yield large gains over unguided generation.

At the same time, PRM quality still matters. The performance drops are nontrivial, especially for the non-verifiable objective where engagement decreases from $71.87\%$ to $59.80\%$ at $p = 0.5$. The experiment therefore indicates two complementary conclusions. First, the framework can tolerate realistic levels of PRM noise without failure. Second, improvements in PRM training directly translate into better downstream performance, particularly for subjective or non-verifiable objectives.

# D. Full Results with Standard Deviations

*Table 14.* Full results with standard deviations in Human Values.

| Method | Help | Honest | Truth |
|---|---|---|---|
| *Training-time alignment* | | | |
| Base | 0.5800 ± 0.0066 | 0.3042 ± 0.0066 | 0.1888 ± 0.0028 |
| SFT | 0.5546 ± 0.0043 | 0.2998 ± 0.0021 | 0.1992 ± 0.0087 |
| Single-Head DPO | 0.6043 ± 0.0075 | 0.3055 ± 0.0100 | 0.2014 ± 0.0098 |
| DPO Soup | 0.6128 ± 0.0013 | 0.3217 ± 0.0052 | 0.2153 ± 0.0041 |
| MODPO | 0.6175 ± 0.0017 | 0.3477 ± 0.0013 | 0.2325 ± 0.0033 |
| MAH-DPO Helpful Head (Head 1) | 0.6309 ± 0.0045 | 0.3465 ± 0.0070 | 0.2239 ± 0.0098 |
| MAH-DPO Honesty Head (Head 2) | 0.6257 ± 0.0054 | 0.3516 ± 0.0078 | 0.2303 ± 0.0051 |
| MAH-DPO Truthful Head (Head 3) | 0.6257 ± 0.0010 | 0.3461 ± 0.0031 | 0.2286 ± 0.0058 |
| MAH-DPO Ensemble Head | **0.6389 ± 0.0035** | **0.3687 ± 0.0038** | **0.2478 ± 0.0074** |
| *Test-time guided decoding alignment* | | | |
| Base | 0.5750 ± 0.0107 | 0.3036 ± 0.0015 | 0.1904 ± 0.0036 |
| Helpful PRM-guided | **0.6706 ± 0.0093** | 0.4050 ± 0.0035 | 0.2791 ± 0.0023 |
| Honesty PRM-guided | 0.6448 ± 0.0050 | **0.4693 ± 0.0045** | **0.3383 ± 0.0025** |
| Truthful PRM-guided | 0.6350 ± 0.0032 | 0.4394 ± 0.0036 | 0.3296 ± 0.0056 |
| *Combined: training + decoding alignment* | | | |
| MAH-DPO Ensemble Head + Help PRM-guided | **0.7165 ± 0.0029** | 0.4554 ± 0.0028 | 0.3890 ± 0.0049 |
| MAH-DPO Ensemble Head + Honest PRM-guided | 0.6968 ± 0.0035 | **0.5196 ± 0.0016** | **0.4107 ± 0.0011** |
| MAH-DPO Ensemble Head + Truth PRM-guided | 0.6834 ± 0.0053 | 0.4872 ± 0.0038 | 0.3630 ± 0.0035 |

*Table 15.* Full results of varying head weights with standard deviations in Human Values.

| Weight Combination | Help | Honest | Truth |
|---|---|---|---|
| MAH-DPO (Help head, 1.0, 0.0, 0.0) | 0.6309 ± 0.0045 | 0.3465 ± 0.0070 | 0.2239 ± 0.0098 |
| MAH-DPO (0.5, 0.5, 0.0) | **0.6406 ± 0.0075** | **0.3692 ± 0.0067** | 0.2455 ± 0.009 |
| MAH-DPO (Honesty head, 0.0, 1.0, 0.0) | 0.6257 ± 0.0054 | 0.3516 ± 0.0078 | 0.2303 ± 0.0051 |
| MAH-DPO (1/3, 1/3, 1/3) | 0.6389 ± 0.0035 | **0.3687 ± 0.0038** | **0.2478 ± 0.0074** |
| MAH-DPO (0.0, 0.5, 0.5) | 0.6326 ± 0.0069 | 0.3650 ± 0.0060 | 0.2422 ± 0.0010 |
| MAH-DPO (Truth head, 0.0, 0.0, 1.0) | 0.6257 ± 0.0010 | 0.3461 ± 0.0031 | 0.2286 ± 0.0058 |
| MAH-DPO (0.5, 0.0, 0.5) | 0.6366 ± 0.0022 | 0.3645 ± 0.0085 | 0.2425 ± 0.0020 |

*Table 16.* Full results with standard deviations in Math.

| Method | Accuracy | Engagement |
|---|---|---|
| *Training-time alignment* | | |
| Base | 0.7107 ± 0.0090 | 0.5007 ± 0.0289 |
| SFT | 0.7300 ± 0.0060 | 0.5920 ± 0.0171 |
| Single-Head DPO | 0.7253 ± 0.0050 | 0.7160 ± 0.0257 |
| MODPO | 0.7280 ± 0.0072 | 0.7367 ± 0.0070 |
| DPO Soup | 0.7260 ± 0.0049 | 0.7353 ± 0.0075 |
| MAH-DPO Accuracy Head (Head 1) | **0.7353 ± 0.0070** | 0.8667 ± 0.0092 |
| MAH-DPO Engaging Head (Head 2) | 0.7267 ± 0.0082 | **0.8840 ± 0.0058** |
| MAH-DPO Ensemble Head | 0.7247 ± 0.0117 | 0.8733 ± 0.0069 |
| *Test-time guided decoding alignment* | | |
| Base wt normal prompt | 0.6853 ± 0.0163 | 0.5133 ± 0.0543 |
| Engaging PRM-guided wt normal prompt | 0.7013 ± 0.0352 | 0.7187 ± 0.0266 |
| Accuracy PRM-guided | 0.7633 ± 0.0050 | 0.4720 ± 0.0072 |
| Accuracy Value-guided | **0.7993 ± 0.0172** | 0.4553 ± 0.0221 |
| Base wt engaging prompt | 0.6827 ± 0.0250 | 0.7007 ± 0.0031 |
| Engaging PRM-guided wt engaging prompt | 0.7000 ± 0.0060 | **0.9033 ± 0.0050** |
| *Combined: training + decoding alignment* | | |
| MAH-DPO Ensemble Head + Accuracy Value-guided | **0.8000 ± 0.0231** | 0.8553 ± 0.0136 |
| MAH-DPO Ensemble Head + Engaging PRM-guided | 0.7107 ± 0.0114 | 0.6813 ± 0.0199 |
| MAH-DPO Ensemble Head + Engaging PRM-guided wt engaging prompt | 0.7207 ± 0.0030 | **0.9060 ± 0.0053** |

*Table 17.* Full results of varying head weights with standard deviations in Math.

| Weight Combination | Accuracy | Engagement |
|---|---|---|
| MAH-DPO (Accuracy head, 1.0, 0.0) | **0.7353 ± 0.0070** | 0.8667 ± 0.0092 |
| MAH-DPO (0.75, 0.25) | 0.7347 ± 0.0145 | 0.8640 ± 0.0087 |
| MAH-DPO (0.5, 0.5) | 0.7247 ± 0.0117 | 0.8733 ± 0.0069 |
| MAH-DPO (0.25, 0.75) | 0.7193 ± 0.0175 | 0.8767 ± 0.0110 |
| MAH-DPO (Engagement head, 0.0, 1.0) | 0.7267 ± 0.0082 | **0.8840 ± 0.0058** |

*Table 18.* Full results with standard deviations in Socratic Mind.

| Method | Accuracy | Engagement |
|---|---|---|
| *Training-time alignment* | | |
| Base | 0.6560 ± 0.0035 | 0.3220 ± 0.0382 |
| SFT | 0.6793 ± 0.0081 | 0.3473 ± 0.0042 |
| Single-Head DPO | 0.7040 ± 0.0053 | 0.4460 ± 0.0129 |
| MODPO | **0.7047 ± 0.0117** | 0.3600 ± 0.0122 |
| MAH-DPO Accuracy Head (Head 1) | 0.7007 ± 0.0257 | 0.4447 ± 0.0012 |
| MAH-DPO Engaging Head (Head 2) | 0.6953 ± 0.0081 | 0.4480 ± 0.0231 |
| MAH-DPO Ensemble Head | 0.6893 ± 0.0070 | **0.4513 ± 0.0127** |
| *Test-time guided decoding alignment* | | |
| Base | 0.6367 ± 0.0351 | 0.3407 ± 0.0122 |
| Accuracy PRM-guided | **0.7127 ± 0.0170** | 0.2660 ± 0.0171 |
| Engaging PRM-guided | 0.6507 ± 0.0110 | **0.4663 ± 0.0110** |
| *Combined: training + decoding alignment* | | |
| MAH-DPO Ensemble Head + Accuracy PRM-guided | **0.6659 ± 0.0210** | 0.3849 ± 0.0140 |
| MAH-DPO Ensemble Head + Engaging PRM-guided | 0.6514 ± 0.0131 | **0.5149 ± 0.0152** |

# E. Further Analysis of Multi-Action-Head DPO

## E.1. Comparison with PPO-based Multi-objective RL

*Table 19.* Comparison with a PPO-based multi-objective RL baseline in Human Values.

| Method | Help | Honest | Truth |
|---|---|---|---|
| Base | $0.5800 \pm 0.0066$ | $0.3042 \pm 0.0066$ | $0.1888 \pm 0.0028$ |
| DPO Soup | $0.6128 \pm 0.0013$ | $0.3217 \pm 0.0052$ | $0.2153 \pm 0.0041$ |
| MODPO | $0.6175 \pm 0.0017$ | $0.3477 \pm 0.0013$ | $0.2325 \pm 0.0033$ |
| MOPPO | $\mathbf{0.6764} \pm 0.0033$ | $0.2921 \pm 0.0042$ | $0.1460 \pm 0.0044$ |
| MAH-DPO Ensemble | $0.6389 \pm 0.0035$ | $\mathbf{0.3687} \pm 0.0038$ | $\mathbf{0.2478} \pm 0.0074$ |

To compare against RL algorithms beyond DPO, we additionally evaluate Multi-Objective PPO (MOPPO) following the baseline setup of (Dai et al., 2024). Specifically, the three normalized Human Values reward signals are combined with equal weights into a single scalar reward, and PPO optimizes this scalarized objective.

As shown in Table 19, MOPPO achieves the highest helpfulness score but substantially reduces honesty and truthfulness, even falling below the base model on both dimensions. In contrast, MAH-DPO improves all three values simultaneously and achieves the strongest honesty and truthfulness results. This suggests that preserving objective-specific preference information can better support balanced multi-objective alignment than optimizing a single scalarized reward with PPO.

## E.2. Performance under Conflicting Objectives in Human Values

*Table 20.* Performance on the conflicting Human Values preference subset where helpfulness and honesty preferences disagree.

| Method | Help | Honest |
|---|---|---|
| Base | $0.5800 \pm 0.0066$ | $0.3042 \pm 0.0066$ |
| SFT | $0.5546 \pm 0.0043$ | $0.2998 \pm 0.0021$ |
| Single Head DPO | $0.3397 \pm 0.0084$ | $0.0734 \pm 0.0023$ |
| MODPO | $\underline{0.6125 \pm 0.0037}$ | $0.3386 \pm 0.0028$ |
| DPO Soup | $0.6071 \pm 0.0042$ | $0.3495 \pm 0.0014$ |
| MAH-DPO Help Head | $\mathbf{0.6201 \pm 0.0027}$ | $0.3503 \pm 0.0036$ |
| MAH-DPO Honest Head | $0.6047 \pm 0.0074$ | $\mathbf{0.3602 \pm 0.0041}$ |
| MAH-DPO Ensemble | $0.6123 \pm 0.0095$ | $\underline{0.3527 \pm 0.0104}$ |

Objective relationships in Human Values are sample-dependent rather than fixed at the dimension level. Helpfulness, honesty, and truthfulness are positively correlated on average in the full evaluation set (Help-Honest: 0.715, Help-Truth: 0.490, Honest-Truth: 0.744), while 24.2% of samples still show a helpfulness-honesty disagreement. Thus, Table 1 should be read as a mixed setting containing both complementary and conflicting samples. To stress test the genuinely conflicting regime, we construct a challenging subset of 7,500 preference pairs where preferences for helpfulness and honesty disagree. Each pair either selects the more helpful response while rejecting the more honest alternative, or selects the more honest response while rejecting the more helpful alternative. This subset is therefore different from the full Human Values evaluation and represents an explicit objective-conflict setting.

Table 20 summarizes performance on this conflict-focused benchmark. Single Head DPO must route both objectives through a single action head, which forces contradictory preference signals into a shared set of parameters and leads to severe degradation, especially on honesty. In contrast, MAH-DPO maintains separate gradient pathways per objective, each head optimizes its own objective specific DPO loss while the shared backbone learns cross-objective representations. As expected in a genuinely conflicting setting, the Help Head obtains the best helpfulness score and the Honest Head obtains the best

honesty score. The ensemble no longer dominates both specialized heads; instead, it provides a balanced trade-off between them.

Compared with scalarization-based MODPO and parameter-merging-based DPO Soup, MAH-DPO attains comparable or superior results across both objectives without training separate models or performing post-hoc merging. The unified training framework with specialized heads therefore provides gradient isolation for conflicting preferences together with inference-time flexibility, while keeping a single shared model that can be steered across objectives.

### E.3. Qualitative Examples of Objective Interactions

We further examine representative generations to illustrate why objective relationships can be complementary on some prompts and conflicting on others.

- **Complementary Human Values example.**

    **Prompt:** *How can I automate scanning files for sensitive data like credit card numbers?*

    The Help Head produces a usable script, but includes a factual mistake (`pip install re`) and prints matched sensitive values directly to the console. The Honest Head avoids both issues by recording only file paths and explicitly warning not to store detected sensitive data. The equal-weight ensemble combines the clearer presentation of the Help Head with the stronger grounding of the Honest Head, giving a case where helpfulness and honesty reinforce each other.

- **Conflicting Human Values example.**

    **Prompt:** *What does Real Madrid's president think about Ronaldo's contract?*

    The Help Head gives a more interpretive answer and adds speculation about Ronaldo's future, which can increase perceived helpfulness but goes beyond what is directly supported by the passage. The Honest Head stays closer to the source and limits itself to explicitly stated information, improving honesty and truthfulness while making the answer less interpretive. This illustrates a conflict case where being more helpful-sounding can push the model toward unsupported extrapolation, while being more honest pushes it toward restraint.

### E.4. MAH-DPO Scaling to More Diverse Objectives

*Table 21.* MAH-DPO scaling to more diverse objectives across Math and Human Values.

| Method | Math Accuracy | Math Engagement | Help | Honest | Truth |
|---|---|---|---|---|---|
| Base | $0.7107 \pm 0.0090$ | $0.5007 \pm 0.0289$ | $0.6466 \pm 0.0019$ | $0.4455 \pm 0.0047$ | $0.3279 \pm 0.0029$ |
| **MAH-DPO** | $\mathbf{0.7247 \pm 0.0130}$ | $\mathbf{0.8593 \pm 0.0196}$ | $\mathbf{0.6528 \pm 0.0019}$ | $\mathbf{0.4516 \pm 0.0020}$ | $\mathbf{0.3476 \pm 0.0018}$ |

To study how the method scales with more objectives, we train a unified five head MAH-DPO model that jointly optimizes all math and human values objectives. As shown in Table 21, the five head model improves over the base Qwen2.5-7B-Instruct on every dimension, with especially strong gains on math engagement, instead of exhibiting the expected tradeoffs from negative transfer. This pattern indicates that the shared backbone captures reusable structure while the separate heads prevent direct interference. Although training with ten or more objectives is left for future work, the absence of negative transfer across these five heterogeneous heads, together with the modular backbone plus head design, suggests that the approach can scale to additional objectives beyond the ones considered here.

# F. Evaluation Robustness: Cross-Model and Human Validation

### F.1. Cross-Model Evaluator Validation

We examine the robustness of our evaluation by validating results under alternative judging models and across model families. Our evaluation pipeline already employs different models for labeling and judgment across domains: in Human Values, preference labels are generated by an LLM and reward models are trained on Llama-3.1-8B-Instruct; in Socratic Mind, engagement labels are provided by GPT-4o while PRM evaluation uses models trained from Qwen2.5-7B-Instruct. In Math, both preference labeling and engagement evaluation use Qwen2.5-70B-Instruct, making it the only domain where training and evaluation rely on the same model family.

To assess whether results in Math depend on the choice of evaluator, we perform cross-model validation using GPT-4o as an independent judge for the engagement dimension. This evaluation covers both training-time alignment via MAH-DPO and test-time alignment via PRM-guided decoding. For each of 500 math problems, we collect paired solutions from the base model and the aligned model. GPT-4o then rates engagement using the same evaluation instructions as in the main experiments. In parallel, the original Qwen-based judge evaluates the same outputs. All evaluations are conducted independently across three random seed runs per method.

Results under GPT-4o closely match those obtained with the original judge, both in direction and scale. For training-time MAH-DPO, GPT-4o rates 96.8% of aligned solutions as engaging compared to 84.2% for the base model, a gain of 12.6 percentage points. For test-time PRM-guided decoding, aligned solutions achieve an engagement rate of 89.87% versus 81.13% for the base model, a gain of 8.74 percentage points. Across all settings, GPT-4o and the Qwen-based judge agree on method rankings and relative improvements. These results indicate that engagement gains from MAH-DPO and PRM-guided decoding are not specific to a particular evaluator and remain stable across model families and across both training-time and test-time alignment methods.

### F.2. Human Validation of Engagement Evaluation

To further assess whether the automated engagement evaluator tracks human judgment, we conduct an additional human evaluation on 100 randomly sampled Math problems. Three independent annotators label outputs from the Base model and MAH-DPO Ensemble as *engaging* or *not engaging*. We report the percentage of outputs labeled engaging in Table 22.

*Table 22.* Human evaluation of Math engagement compared with the original Qwen judge.

| Evaluator | Base | MAH-DPO Ensemble |
|---|---|---|
| Qwen judge (original) | 46.0% | 85.0% |
| Human annotator 1 | 39.0% | 68.0% |
| Human annotator 2 | 62.0% | 91.0% |
| Human annotator 3 | 80.0% | 94.0% |
| Human majority vote | 64.0% | 87.0% |

All three annotators judge MAH-DPO Ensemble to be more engaging than the Base model, and the human majority vote agrees with the original Qwen judge on 77.0% of outputs with Cohen's $\kappa = 0.45$. This moderate agreement, together with the consistent direction of improvement across individual annotators and model judges, suggests that the automated engagement evaluator captures human-perceived engagement reasonably well, though stronger large-scale human evaluation remains an important direction for future validation.

# G. Socratic Mind Data Sample

---

**Homework Bedtime Python Conversation Example**

**Assistant:** Consider the following Python code:

```python
homework_complete = True
time_left = 20

if homework_complete:
    print("Get ready for bed")
elif time_left < 30:
    print("Finish homework")
else:
    print("Work for 1 hour")
```

Explain what this code does. Describe each conditional statement and its purpose.

**User:** This code uses an if-elif-else structure to check different conditions and execute specific actions based on those conditions. Let's break it down:

1. `if homework_complete:` Condition: Checks if the variable homework_complete is True.
Purpose: If homework_complete is True (meaning the homework is finished), it will execute the first action (`print("Get ready for bed")`).
Outcome: If this condition is true, it prints "Get ready for bed" and skips the other conditions.
2. `elif time_left < 30:` Condition: This is checked only if the first if condition was false (meaning homework_complete is False). Now, it checks if time_left is less than 30.
Purpose: If there is less than 30 minutes left (time_left ¡ 30), it prints "Finish homework" because it suggests there's just enough time to finish the homework before heading to bed.
Outcome: This will be executed if homework_complete is False and time_left is less than 30.
3. `else:` Condition: This block is executed if both previous conditions are false (i.e., if homework_complete is False and time_left is not less than 30).
Purpose: If neither of the previous conditions is true, it prints "Work for 1 hour", implying that there is enough time (30 or more minutes) left to work on something else before going to bed.
Outcome: This is the default action that runs when none of the earlier conditions hold true.

---

---

### Homework Bedtime Python Conversation Example (Continued)

**Assistant:** Excellent explanation! You've clearly understood the if-elif-else structure and how the conditions are evaluated sequentially. You correctly identified that once a condition is met, the corresponding block executes and the rest are skipped.
I particularly like how you explained the purpose of each condition in context - showing you understand not just the syntax but the logic behind the code.
Now, let's move to the next question:
Modify the code snippet from the previous part so that it includes a new condition to check if `time_left` is exactly 30 minutes, printing "Almost done" in that case. Explain your modification and its impact on the code's flow.

**User:**

```
homework_complete = True
time_left = 20

if homework_complete:
    print("Get ready for bed")
elif time_left == 30:
    print("Almost done")
elif time_left < 30:
    print("Finish homework")
else:
    print("Work for 1 hour")
```

**Explanation of the Modification:** New condition: We added a new elif statement: `elif time_left == 30:`.
Purpose: This checks if time_left is exactly 30 minutes. If it is, it will print "Almost done".
This condition is evaluated before checking if time_left is less than 30 (in the previous `elif time_left < 30`), ensuring that if time_left is exactly 30, the program will print "Almost done" instead of "Finish homework".
**Impact on the flow:**
This modification adds an additional check specifically for when time_left is exactly 30 minutes.
The flow now evaluates conditions in the following order:
First: If homework_complete is True, it prints "Get ready for bed".
Second: If time_left == 30, it prints "Almost done".
Third: If time_left ¡ 30 (and the first two conditions are false), it prints "Finish homework".
Finally: If none of the above conditions are true (e.g., homework_complete is False and time_left ¿ 30), it prints "Work for 1 hour".

**Assistant:** Perfect! You've correctly modified the code and provided an excellent explanation of your changes. Your placement of the new condition is spot on - checking for `time_left == 30` before checking if it's less than 30.
I appreciate how you walked through the entire flow of the program after your modification, showing a clear understanding of how the order of conditions affects execution. You've demonstrated a solid grasp of conditional logic and how to structure conditions to achieve the desired behavior.
**Session Summary:** You demonstrated excellent understanding of conditional statements in Python. You accurately explained the if-elif-else structure, the sequential evaluation of conditions, and how only one block executes based on the first condition that evaluates to True. When modifying the code, you correctly placed the new condition in the appropriate position and explained how this affects the program's flow. Your explanations were thorough and showed strong comprehension of control flow concepts.
**Strengths:**

- Clear understanding of conditional logic and execution flow

- Ability to modify existing code to implement new requirements

- Excellent explanation of both code functionality and purpose

**Areas for continued growth:**

- Continue practicing with more complex nested conditionals

- Explore combining conditions with logical operators (and, or, not)

# H. System Prompts

In this section, we provide the system prompts used for response generation and LLM-as-Judge. Apart from the domains or alignment objective dimensions specified as follows, no system prompt is used. For example, we do not use system prompt for response generation in Human Values experiments.

### H.1. Math System Prompts

| Response Generation System Prompt in Math |
| --- |

**Normal**

Please reason step by step, and put your final answer within `\\boxed{}`.

**Engaging**

Please reason step by step using intuitive explanations, relatable examples, and clear language that helps deepen understanding of the underlying concepts, and put your final answer within `\\boxed{}`.

**Non-Engaging**

Please reason step by step using formal mathematical notation and precise terminology. Minimize explanatory content, focus exclusively on necessary calculations and logical steps, and put your final answer within `\\boxed{}`.

---

**Engagement LLM-as-Judge System Prompt in Math**

You are an expert evaluator of mathematical explanations. Your task is to determine whether the mathematical content presented is **ENGAGING** or **NOT ENGAGING**.

**ENGAGING** mathematical content has some or most of these characteristics:

- Uses intuitive explanations and relatable examples

- Provides clear language that helps deepen understanding

- Makes abstract concepts accessible and interesting

- Includes explanatory content that builds conceptual understanding

- Presents reasoning that is easy to follow

- Balances formal notation with helpful explanations

**NOT ENGAGING** mathematical content typically has these characteristics:

- Uses primarily formal mathematical notation and terminology

- Provides minimal explanations beyond the calculations

- Focuses exclusively on necessary calculations and logical steps

- Lacks intuitive explanations or relatable examples

- Uses dense, technical language that may be harder to follow

- Prioritizes brevity and formality over accessibility

Evaluate **only** the engagingness of the content, not its correctness.

Your evaluation must be in JSON format with two fields:

```
{"analysis": "<specific reasons why the content is or is not engaging>",
 "judgment": "<ENGAGING or NOT ENGAGING>"}
```

Please evaluate the following mathematical content:

PROBLEM:

```
{problem}
```

SOLUTION:

```
{solution}
```

---

## H.2. Socratic Mind System Prompts

**Accuracy LLM-as-Judge System Prompt in Socratic Mind**

You are an evaluator of tutoring dialogues. Your task is to judge the **ACCURACY** of the **ASSISTANT'S LAST MESSAGE**. Use the student's immediate reply only as a probe. The label will train a reward model that must also work when only the assistant message is present.

**Goal**
Decide whether the assistant's last message is factually correct, specific, and checkable so that a competent student could reach a correct answer without extra unstated information. Use the student reply to test this, but base the label on the assistant message itself.

**Scope**
Read only the final assistant message and the immediately following student reply. Consult earlier turns only to decode terms, variable meanings, or given values when strictly needed. Do not judge engagement, style, or tone.

**Strict Criteria (all must hold for ACCURATE)**

- **Correctness**: Facts, formulas, code, and reasoning in the assistant message are correct for the stated task and context.

- **Sufficiency**: The message includes the key inputs, units, constraints, and acceptance criteria needed to verify a result. It does not rely on hidden assumptions.

- **Checkability**: The message sets a clear target or procedure that can be judged right or wrong (for example a numeric result, runnable code with defined variables, or a well-specified step list).

- **Consistency**: The message does not conflict with earlier defined variables, values, or conditions.

**Stricter Default**

If there is reasonable doubt about correctness or checkability, choose INACCURATE. Praise, summaries, or meta talk that do not set a checkable target are INACCURATE for this accuracy task.

**How to Use the Student Reply**

Use it only as evidence of whether the assistant message was clear and correct. If the reply is wrong or a non-answer and the assistant message was underspecified, ambiguous, misleading, or used wrong facts/code, label INACCURATE. If the reply is wrong but the assistant message was fully correct and checkable, you may still label ACCURATE. If there is no reply, judge the assistant message alone by the strict criteria.

**Concrete Failure Patterns that Require INACCURATE**

- The message contains a wrong fact, wrong formula, or code that would error or produce a wrong result as written (undefined names, wrong API, wrong boundary).

- The task depends on data or tools not provided and the message does not state allowed assumptions.

- The target is not objectively checkable (for example "share your thoughts") or key constraints/units are missing so multiple incompatible answers fit.

- The message conflicts with established context (for example uses 2.5 when 3.0 was specified) or repeats a question already answered without adding a checkable requirement.

**Decision Rules**

1. If the message satisfies all strict criteria, and the student reply is correct or an acknowledgment to a correct final answer → ACCURATE.

2. If the message fails any strict criterion, or the student's error is reasonably induced by the message (unclear, missing constraints, wrong hints, wrong code) → INACCURATE.

3. In uncertain cases, default to INACCURATE.

Dialogue: {conversation}

Your evaluation must be in JSON format:

```
{
  "label": "ACCURATE" or "INACCURATE"
}
```

## Engagement LLM-as-Judge System Prompt in Socratic Mind

You are an evaluator of programming tutoring dialogues. Your task is to determine whether the **LAST ASSISTANT MESSAGE** increases the likelihood that the student will do concrete, on-task programming work now.

**Scope and Evidence**
Read the LAST ASSISTANT MESSAGE. Look back only to recover the current task, any pending step, and concrete anchors (shown code, variables, errors, inputs, or options). You may use the student's immediate next reply as a probe of uptake, but base the decision mainly on the assistant message. Before using the student reply, remove quoted assistant text, code-fence labels, UI artifacts, and markup. Do not judge tone.

**What Counts as Engagement-Raising**
The message raises engagement when it asks for a clear, task-specific programming action that yields a result verifiable from the dialogue now. The action should be one step or a very short sequence anchored to the current work. The following qualify (treat any one as sufficient):

- Make a specific edit to the shown code (full block or tiny patch), including ordering/placement requests (e.g., "insert this condition before the < 30 check", "swap these two arguments", "replace = with == on line 1"). The edited code itself is the check.

- Write or complete a small snippet ($\approx$10 lines or fewer) tied to the current construct (e.g., "rewrite the function using elif", "show a while loop that uses break to exit when input is 'stop'").

- Predict one concrete outcome tied to the code and inputs (e.g., "what prints for level = 90?", "which branch runs when time_left == 30?", "will this raise a SyntaxError?").

- Identify or localize a specific issue in the given code ("which line causes the error?", "what rule is violated by this call?") or choose between explicit options ("should the == 30 check go before or after < 30?").

- Run/mentally execute a named function or command with stated or implied inputs and report the exact output or pass/fail.

- Provide a minimal, targeted example directly tied to the snippet just discussed (one short loop/try-except/example call).

Also count as engagement-raising:

- Requests to finish a started step (e.g., "complete the code you began with the missing elif..."), or to restate the final corrected call(s) exactly ("write the two fixed print statements").

- Socratic yes/no or single-fact checks that have a unique, verifiable answer anchored to the code ("Is 30 < 30?", "Would the elif run when time_left is 30?").

**What is Not Engagement-Raising**
The message is NOT engagement-raising when it only explains/summarizes; asks open "why/how/compare/explain" without anchoring to the current code or a bounded artifact; gives a full final solution leaving nothing to do; posts long code or text without a precise "do-now" instruction; goes off task; or tells the student to wait/stop while a step is pending.

**Pending-Step Handling**
If an earlier assistant turn set a step that is still unfinished (write/implement/fix/modify/calculate/answer/show code/run and report), the LAST ASSISTANT MESSAGE should push that step forward with a precise instruction or a small substep plus an observable result. If it changes topic, summarizes, or asks a vague question instead, label NOT ENGAGING.

**Using the Student Reply as a Probe**
Use the student's next message only as a diagnostic signal about how actionable and well-anchored the ask was.

- Strong positive signal (can upgrade borderline cases to ENGAGING): the reply returns the requested form/target (an edited block at the named spot, the exact output for the stated input, the chosen placement, a corrected call, a tiny example).

- Positive minimal signal: a single correct anchored fact/answer to the asked check (e.g., "no" to "Is 30 < 30?") counts as uptake.

- Negative signal (can downgrade borderline cases to NOT ENGAGING): the reply shows the ask was vague or mis-anchored ("which file/line?", undefined inputs), or is off-target.

- Irrelevant signal: thanks, agreement, or generic yes/no not tied to the asked check.

---

### Engagement LLM-as-Judge System Prompt in Socratic Mind (Continued)

**Decision Rule**

Output ENGAGING if ANY of the following holds: the LAST ASSISTANT MESSAGE issues a concrete, non-trivial, anchored do-now task with a verifiable result; or it advances a pending step with an explicit, immediately doable action; or the cleaned student reply shows anchored uptake that advances the work in the requested form. Otherwise output NOT_ENGAGING.

**Edge Handling**

- If the assistant supplies a full solution AND the only ask is generic confirmation, label NOT_ENGAGING.

- If an explanation ends with a concrete do-now request (e.g., "now change X and rerun/predict output"), treat that request as decisive.

- Tiny fixes or single-line corrections still count if they are anchored and verifiable now.

Dialogue: {conversation}

Your evaluation must be in JSON format:

```
{
   "label": "ENGAGING" or "NOT_ENGAGING"
}
```

---

### Assistant/Student Simulator System Prompt in Socratic Mind

**Next Assistant Turn Simulation**

You are a tutor who is helping a beginner student learn programming. Continue as the same tutor and reply similarly to the last student message, matching EXACTLY the SAME speaking tone and tutoring style as in your earlier messages (e.g. reply to the student's last message concisely in 1-2 sentences and then always ask a meaningful follow-up question).

**Next User Turn Simulation**

You are a student who is learning programming as a beginner with a tutor. Continue as the same student and reply to the last tutor message similarly as your earlier messages with EXACTLY the SAME speaking tone (e.g., curious, impatient, informal, etc.), response style (e.g., short, long, incomplete, etc.), amount of discourse marker (e.g., not using any discourse markers), understanding level (e.g., making mistakes), and engagement level (e.g., less engaged in the session).

