# OpenReview forum: "Simultaneous Multi-objective Alignment Across Verifiable and Non-verifiable Rewards"
_ICML.cc/2026/Conference — ICML 2026 regular_

### Official Review · Reviewer_7hxX · 2026-03-07

**Soundness:** 2
**Presentation:** 3
**Significance:** 3
**Originality:** 3
**Overall Recommendation:** 4
**Confidence:** 4

**Summary:**

- In the real world, objectives are often multi-dimensional; for example, satisfying human engagement or preferences while simultaneously ensuring task accuracy.
- This paper proposes learning a multi-action head to the shared base LLM using multi-objective DPO (i.e., H preference datasets for H objectives, where each head is specialized for a unique preference or reward), and at test-time to balance different objectives, the decoding process takes in individualized weights $w_i$.
- Experiments with Math, Human Values, and Socratic Method are implemented with Qwen 7B instruct models and show empirical advantages compared to baselines and other multi-objective DPO methods.

**Compliance With Llm Reviewing Policy:**

Affirmed.

**Final Justification:**

I appreciate the author's rebuttal and my initial concern is resolved via the author's second rebuttal.

**Key Questions For Authors:**

- Please review the weakness section; if these questions are answered adequately and the confusion about conflicting / complimentary objectives is resolved with further experiments or writing edits, this reviewer is willing to change the score.

**Limitations:**

Yes

**Strengths And Weaknesses:**

- Strength:
- Tunable weights for each objective head allow for balancing of different (potentially conflicting) objectives at inference time, without additional finetuning.
- This method could potentially be used for pluralistic alignment, where different users exhibit different, potentially conflicting, preferences without needing to additionally finetune the model for each unique preference. (Therefore citing works in pluralistic preference alignment would be useful, for example: see Sorensen et al., 2024 "A Roadmap to Pluralistic alignment" https://arxiv.org/abs/2402.05070 and Kirk et al., 2024 "The PRISM alignment dataset" https://arxiv.org/abs/2402.05070).

- Weaknesses:
- Regarding Section 4.2 Case C, it's unclear how a reward model trained only on complete answers can estimate partial correctness or provide a good proxy for partial process rewards.
- This paper doesn't clearly distinguish between cases where (1) multi-objectives complement each other (for example, increasing the user engagement and the model's problem solving accuracy do not seem like conflicting objectives, and therefore we'd expect a model that's optimized for one objective to not affect its performance on the other objective); and (2) objectives that conflict with each other (for example, in Human Values dataset, users who prefer helpfulness may select response A over B while users who prefer honesty may select B over A, so in such cases, optimizing for one preference / objective would be expected to unavoidably degrade performance in the other metric). Making this distinction would help provide more clarity to the experiment results. For example, it's unclear how in Table 1, human values, MAH-DPO ensemble is able to obtain the best performance in all three metrics, and even better than individual preference heads (like MAH-DPO Help Head, MAH-DPO Honest Head etc.) -- if at inference time, different weights are applied to MAH-DPO Ensemble based on whether the prompt requires specializing in helpfulness over honesty etc., this needs to be clarified to avoid confusion.
- In that regard, we wouldn't expect performance to drop for Math and Socratic depending on whether the objective focuses on accuracy versus engagement, because these do not reflect two conflicting objectives; however, Table 1 seems to suggest that there are empirical trade-offs observed for MAH-DPO ensemble when optimizing for Acc versus Engagement (sub tables a and c).
- If authors believe that there are reasons or justifications for the performance trade-offs (i.e., optimizing for accuracy during decoding results in reduced engagement, vice versa; providing qualitative dialogue examples of when and how this would happen would be helpful.) Similarly, for Human Values, where we would expect responses optimized for Helpfulness to be qualitatively different from responses optimized for Honesty, it would be useful to provide qualitative analysis of the generated responses that demonstrate these conflicting values. Alternatively, the authors could also consider clustering or visualizing response embeddings by MAH-DPO from each preference head to further showcase that the responses optimized for Helpfulness are different from those optimized for Truthfulness; while responses generated by SFT or MODPO cannot distinguish different preference targets because their decoding process doesn't have a way of upweighting one specific dimension over others.
- Nit formatting fix: Table 2, MAH-DPO Help is boldfaced even though MODPO performs better (0.6125).

---

> ### Author Rebuttal · Authors · 2026-03-31
>
> Thank you for your insightful and valuable feedback, and we address each point below.
>
> **Response to Weakness 1:**
>
> Thanks for raising this point. Case C is not meant to verify exact local partial correctness. It is designed for settings such as Human Values, where no ground-truth stepwise correctness exists, and uses a partial response as a proxy for expected final response quality during decoding. In our implementation, this proxy is only applied to semantically coherent prefixes, e.g., at sentence terminators and paragraph breaks. This is consistent with prior works, e.g., [1], which suggests that reward models trained on full responses can remain reliable on semantically complete segments.
>
> [1] Cascade Reward Sampling for Efficient Decoding-Time Alignment. COLM 2025
>
> ---
> **Response to Weakness 2-4:**
>
> > This paper does not clearly distinguish between...
>
> Our additional analysis suggests that objective relationships are often sample-dependent rather than fixed at the dimension level: the same pair of objectives may align on one prompt but conflict on another. For example, Human Values is a mixed regime: the objectives are positively correlated on average (Help vs. Honest: 0.715, Help vs. Truth: 0.490, Honest vs. Truth: 0.744), while 24.2\% of samples still show a helpfulness-honesty trade-off. In contrast, Math and Socratic are mostly complementary settings: accuracy and engagement usually improve together, but they are not fully independent because both are expressed through the same generated text.
>
> > In Table 1, it is unclear how MAH-DPO Ensemble…
>
> We clarify that MAH-DPO Ensemble in Table 1b uses fixed equal weights, \($w=(1/3,1/3,1/3)$\), not prompt-adaptive reweighting. The ensemble can outperform the individual heads because on many prompts the three objectives move in the same direction, and thus combining heads preserves shared gains across helpfulness, honesty, and truthfulness. On the conflicting subset, the ensemble reduces overspecialization from any single head and often gives a better average trade-off. This interpretation is also consistent with Table 2, where MAH-DPO remains strong on an explicit helpfulness-honesty conflict subset.
>
> > For Math and Socratic, we would not expect...
>
> This intuition is largely correct, and the observed trade-offs are mild rather than fundamental. In Math, only 7.2\% of problems are solved only by the Accuracy Head and not the Engagement Head. These tend to be cases where engagement-oriented verbosity or exploratory reasoning makes the solution less direct or weakens systematic checking. We will clarify that Math and Socratic are mostly complementary settings, with mild trade-offs because both objectives act on the same output.
>
> > It would be useful to provide qualitative analysis...
>
> We provide representative qualitative examples below.
>
> On `How can I automate scanning files for sensitive data like credit card numbers?`, the Helpful Head produces a usable script, but it includes a factual mistake (“pip install re”) and prints matched sensitive values directly to the console. The Honest Head avoids both issues by recording only file paths and explicitly warning not to store the detected sensitive data. The Ensemble combines the stronger grounding of the Honest Head with the clearer presentation of the Helpful Head, and achieves the best scores across all three dimensions. This is a case where the objectives reinforce each other, so ensembling helps.
>
> On `What does Real Madrid’s president think about Ronaldo’s contract?`, the Helpful Head gives a more interpretive answer and adds speculation such as “this stance could signal that Ronaldo’s future might be uncertain“, which increases helpfulness but goes beyond what is directly supported by the passage. The Honest Head stays closer to the source and limits itself to what the passage explicitly states, which improves honesty and truthfulness but makes the answer less interpretive. This is a conflict case: offering a more helpful-sounding analysis pushes the model toward speculation, while being more honest pushes it toward restraint.
>
> A similar pattern appears in Math. On `How many values can be obtained from \(2 \cdot 3 \cdot 4 \cdot 5 + 1\) by inserting parentheses?`, the Accuracy Head systematically identifies all valid groupings and reaches the correct answer, while the Ensemble explores the space more verbosely and ends up missing key cases. This suggests that engagement-oriented exploration is usually compatible with correctness, but on a small subset of harder examples it can make the reasoning less focused.
>
> Overall, Human Values mixes aligned and conflicting prompts, whereas Math and Socratic are mostly complementary with mild trade-offs. We will clarify the distinction in Section 6, state explicitly that Table 1b uses fixed equal ensemble weights, and add representative qualitative examples.
>
> ---
> **Response to Weakness 5:**
>
> Thank you, and we have fixed the formatting in the revised version.

---

> > ### Author Rebuttal · Reviewer_7hxX · 2026-04-03
> >
> > Thanks for additional clarification, I still have some outstanding question about the performance on the conflicting preference subset.
> >
> > Re conflicting preference subset:
> > I agree with the authors that in the complementary dataset, "[t]he ensemble can outperform the individual heads because on many prompts the three objectives move in the same direction, and thus combining heads preserves shared gains across helpfulness, honesty, and truthfulness."
> >
> > And also agree that "overspecialization" of any one head may be more helpful in terms of **average performance** even on a conflicting dataset. However, specifically regarding Table 1, (b) last row -- if this is tested on the conflicting preference subset -- then it's not clear why the individual head "overspecialized" for helpfulness is outperformed by the jointly optimized head, and similarly for honesty, and helpfulness. My specific confusion is that: If the preferences are truly conflicting (lets say 50% of people prioritizing honesty prefer $y_1$ over $y_2$, while 50% of people prioritizing helpfulness prefer $y_2$ over $y_1$), then how can a model that optimizes for both preferences outperform a model that always chooses $y_1$ (overspecialized on honesty-prioritizing users and therefore always choosing $y_1$) and another model that always chooses $y_2$ (overspecialized on helpfulness)? This makes me wonder whether the dataset includes non-conflicting preferences, or whether the user's preferences are clearly distinguishable from the input prompts, or otherwise how the benefits transfer across conflicting preferences via a joint reward model head.

---

> > > ### Author Response · Authors · 2026-04-04
> > >
> > > Thank you for your reply and for acknowledging that our responses have addressed many of your previous concerns. We appreciate the opportunity to further clarify your remaining confusion:
> > >
> > > > However, specifically regarding Table 1, (b) last row -- if this is tested on the conflicting preference subset -- then it's not clear why the individual head "overspecialized" for helpfulness is outperformed by the jointly optimized head, and similarly for honesty, and helpfulness.
> > >
> > > > This makes me wonder whether the dataset includes non-conflicting preferences, …
> > >
> > > We would like to clarify that the experimental results in Table 1(b) are conducted on the dataset without any preference-specific filtering. Therefore, the dataset includes both conflicting and non-conflicting preferences as you suggested, which explains why head ensemble outperforms individual heads.
> > >
> > > > If the preferences are truly conflicting (lets say 50% of people prioritizing honesty prefer $y_1$ over $y_2$, while 50% of people prioritizing helpfulness prefer $y_2$ over $y_1$), then how can a model that optimizes for both preferences outperform a model that always chooses $y_1$ (overspecialized on honesty-prioritizing users and therefore always choosing $y_1$) and another model that always chooses $y_2$ (overspecialized on helpfulness)?
> > >
> > > That said, the results in Table 2 are conducted on conflicting preference pairs where preferences for helpfulness and honesty disagree (detailed setup is described in Appendix E.1), which is exactly the truly conflicting setting you described. We further report the results of individual heads under this setting as below:
> > >
> > > | Method | Help | Honest |
> > > |--------------------|--------|--------|
> > > | Base (reported in Table 2) | 0.5800 | 0.3042 |
> > > | MAH-DPO Help Head | **0.6201** | 0.3503 |
> > > | MAH-DPO Honest Head | 0.6047 | **0.3602** |
> > > | MAH-DPO Ensemble (reported in Table 2) | 0.6123 | 0.3527 |
> > >
> > > The results confirm that your intuition is correct. Under the preference-conflicting setting, the head ensemble does not outperform individual heads on their own specialized dimensions, but instead achieves a more balanced trade-off across dimensions.
> > >
> > > We will include these clarifications in Section 6.1 regarding the different settings in Table 1(b) and Table 2, and also include the discussion on complementary versus conflicting preferences as you have suggested. Thank you again for your valuable feedback, and we hope our response has effectively addressed your remaining concerns.

---

### Official Review · Reviewer_wgMj · 2026-03-08

**Soundness:** 3
**Presentation:** 3
**Significance:** 3
**Originality:** 2
**Overall Recommendation:** 4
**Confidence:** 4

**Summary:**

This paper addresses the challenge of simultaneous multi-objective alignment for LLMs across verifiable rewards, non-verifiable subjective preferences, and interactive multi-turn scenarios, by proposing a unified training-and-inference framework that avoids collapsing heterogeneous feedback into a single objective. The core approach has three parts: (i) a standardized process reward model (PRM) training pipeline, (ii) Multi-Action-Head DPO (MAH-DPO), and (iii) PRM-guided decoding. Experiments show that across math reasoning, human values, and multi-turn tutoring, the framework can improve multiple objectives jointly with limited interference and provides smooth inference-time control via head weighting and PRM guidance.

**Compliance With Llm Reviewing Policy:**

Affirmed.

**Final Justification:**

My concerns are addressed, so I have raised my score to 4.

**Key Questions For Authors:**

Please refer to the numbered items in the Weaknesses section above. If the authors can properly address my concerns, I am willing to raise my score.

**Limitations:**

The authors do not appear to discuss the limitations of the proposed approach. For my suggestions, please see the Weaknesses section above.

**Strengths And Weaknesses:**

**Strengths:**
1. Simultaneous multi-objective alignment is an important and practical problem.
2. The paper is well presented overall, with clear figures and tables, a logical flow, and helpful formulas.
3. The experimental evaluation is fairly thorough.

**Weaknesses:**

*If the authors could properly address them during the rebuttal phase, I am willing to raise my score.*
1. The proposed framework does not seem to have a single, clear name, even though some individual modules are named. This may hurt recognizability and adoption. I suggest giving the overall framework a concise and memorable name.
2. A key advantage is "low interference across objectives", but the paper lacks discussion of prior work on mitigating conflicts across tasks/objectives (e.g., [1][2][3][4]). Even if the proposed approach is technically different, positioning it relative to these lines of work would significantly strengthen the narrative and clarify what is new.
3. The paper discusses how noise in the math domain affects the method in detail. This raises the question of whether DPO in other domains is similarly sensitive to noise. It would be helpful to discuss relevant noise-robust training work (e.g., [5][6]) and how those ideas relate to this setting.
4. The PRM-guided decoding relies on step segmentation heuristics (e.g., newline boundaries in math; sentence/paragraph breaks in human values), which may be brittle and could affect both quality and compute. The authors may want to discuss the robustness to segmentation errors.
5. The evaluation includes cross-model validation in parts, but the overall pipeline still depends on LLM-based labeling or judging in several places. The authors may want to discuss the evaluator bias and stronger human evaluation (or alternative judges).


[1] How Abilities in Large Language Models are Affected by Supervised Fine-Tuning Data Composition. ACL 2024.

[2] Mixture-of-Skills: Learning to Optimize Data Usage for Fine-Tuning Large Language Models. EMNLP 2024.

[3] Boosting Multi-Domain Fine-Tuning of Large Language Models through Evolving Interactions between Samples. ICML 2025.

[4] Boosting Multi-Domain Reasoning of LLMs via Curvature-Guided Policy Optimization. ICLR 2026.

[5] Provably Robust DPO: Aligning Language Models with Noisy Feedback. ICML 2024.

[6] ROPO: Robust Preference Optimization for Large Language Models. ICML 2025.

---

> ### Author Rebuttal · Authors · 2026-03-31
>
> We appreciate your thorough and helpful feedback. We address your concerns and questions as follows.
>
> **Response to Weakness 1:**
>
> We thank the reviewer for this suggestion. We plan to refer to our full framework as **MAHALO** (**M**ulti-**A**ction-**H**ead **AL**ignment with PRM-guided Dec**O**ding).
>
> ---
> **Response to Weakness 2:**
>
> Thanks for pointing us to these related works. DMT [1], MoS [2], and EVIC [3] mainly mitigate interference through SFT scheduling, data weighting, or sample selection, while CGPO [4] does so through curvature-guided gradient preconditioning across domains during policy optimization. Our method operates through a different mechanism: in MAH-DPO, the low-interference property comes from a shared backbone with objective-specific action heads and per-objective preference loss routing during training. This design also keeps objective trade-offs directly controllable at inference time through head selection or ensembling. We will clarify this positioning in our revised paper.
>
> ---
> **Response to Weakness 3:**
>
> Thanks for the suggestion. Recent work such as rDPO [5] and ROPO [6] shows that vanilla DPO can be sensitive to noisy pairwise preferences beyond math-specific settings. Our Appendix C.4 studies a different noise source, namely noisy process-level guidance at inference time rather than noisy preference labels during DPO training. These directions are complementary, and [5,6] could be incorporated into MAH-DPO when head-specific preference pairs are noisy.
>
> ---
> **Response to Weakness 4:**
>
> Thanks for the suggestion. We conducted additional ablations on the segmentation strategy used in PRM-guided decoding. In Math, we replaced the default `\n\n` boundaries with fixed-interval segmentation every 128 or 64 tokens, which can cut across sentences or equations. In Human Values, the default setup uses both paragraph and sentence boundaries; we ablated this by using only paragraph boundaries or only sentence boundaries.
>
> **Math**
>   | Strategy | Accuracy (%) | Avg. Steps |
>   |:---|:---:|:---:|
>   | `\n\n` (default) | **79.9** | 7.2 |
>   | Fixed every 128 tokens | 77.2 | 4.9 |
>   | Fixed every 64 tokens | 73.0 | 9.3 |
>
> **Human Values**
>   | Strategy | Helpfulness | Honesty | Truthfulness |
>   |:---|:---:|:---:|:---:|
>   | Sent.+para. (default) | **0.671** | **0.405** | **0.279** |
>   | Paragraph only | 0.646 | 0.378 | 0.240 |
>   | Sentence only | 0.662 | 0.374 | 0.253 |
>
> These results suggest that the method is reasonably robust to segmentation choice: the default content-aware heuristics perform best, but simpler or mismatched boundaries lead to graceful rather than abrupt degradation. We will include these ablations in our paper.
>
> ---
> **Response to Weakness 5:**
>
> Thanks for raising this point. As detailed in Appendix F, **Math Engagingness** is the only setting where both preference labeling and engagement evaluation rely on the same model family. In contrast, **Math Accuracy** has verifiable ground truth, **Human Values** uses externally collected UltraFeedback annotations together with trained reward models, and **Socratic Mind** labels are calibrated against real student post-interaction surveys. Appendix F therefore already includes cross-model validation for **Math Engagingness** using GPT-4o as an independent judge to assess evaluator bias.
>
> To fully address evaluator bias, we conducted an additional human evaluation on 100 randomly sampled Math problems, where three independent annotators labeled Base and MAH-DPO Ensemble outputs as ENGAGING or NOT ENGAGING. We report the ENGAGING rate in the table.
>   | Evaluator | Base | MAH-DPO |
>   |---|---|---|
>   | Qwen judge (original) | 46.0% | 85.0% |
>   | Human annotator 1 | 39.0% | 68.0% |
>   | Human annotator 2 | 62.0% | 91.0% |
>   | Human annotator 3 | 80.0% | 94.0% |
>   | Human majority vote | 64.0% | 87.0% |
>
> All three annotators judge MAH-DPO Ensemble to be more engaging than the base model, and the human majority vote agrees with the original Qwen judge on 77.0% of outputs (Cohen’s $\kappa$ = 0.45), suggesting that the automated evaluator tracks human-perceived engagement reasonably well.

---

> > ### Author Rebuttal · Reviewer_wgMj · 2026-04-01
> >
> > Thank you for the careful rebuttal. My concerns have been addressed, so I will raise my score to 4. Good luck.

---

> > > ### Author Response · Authors · 2026-04-04
> > >
> > > Thank you for your careful rebuttal review and for raising your score after considering our response. We are very glad that our rebuttal addressed your concerns, and we sincerely appreciate your time and support.

---

### Official Review · Reviewer_Umiw · 2026-03-09

**Soundness:** 3
**Presentation:** 3
**Significance:** 4
**Originality:** 3
**Overall Recommendation:** 4
**Confidence:** 4

**Summary:**

The authors argue that standard alignment pipelines, such as RLHF or DPO, "collapse" multi-dimensional human preferences into a single scalar objective, which discards valuable trade-off information and limits user control during inference . To solve this, they propose a system that preserves the structured nature of human feedback through three coordinated components:

Standardized PRM Training: A pipeline to derive fine-grained, step-level supervision across different domains.

Multi-Action-Head DPO (MAH-DPO): A training method using a vectorized reward where different dimensions correspond to distinct objectives.

PRM-Guided Decoding: An inference-time optimization that utilizes a "continuing hidden state" to offer precise control of head selection.

**Compliance With Llm Reviewing Policy:**

Affirmed.

**Final Justification:**

My concerns have been addressed. And I think this is a borderline paper, I will maintain my score.

**Key Questions For Authors:**

1. Have you conducted experiments with other related RL algorithms?
2. Have you analyzed the frequency or rationality of individual action heads during the inference phase?

**Limitations:**

While the proposed framework offers a novel approach to multi-objective alignment, several limitations should be noted:

Analysis of Head Selection Dynamics: The paper lacks a detailed empirical analysis regarding the selection frequency and rationality of individual action heads during the inference phase. Specifically, there is no quantitative data showing how often the model switches between.

Granularity of MAH Design: The architectural details of the Multi-Action-Head (MAH) layers could be more finely detailed . A more exhaustive exploration of the specific hyperparameters or structural variations within these heads would provide better guidance for researchers looking to replicate or optimize the design.

Algorithmic Scope: The experimental evaluation is primarily focused on DPO-based alignment. The paper would benefit from additional experiments involving other reinforcement learning paradigms to demonstrate the broader generalizability of the multi-head framework.

**Strengths And Weaknesses:**

Strength:
1. The paper is technically grounded and supported by extensive experimental results.
2. The introduction of MAH-DPO is a creative extension of the DPO algorithm, allowing a single backbone to maintain multiple "personalities" or objectives simultaneously through independent projection heads.
3. The logical synergy between the training and inference phases is a standout strength of this work, significantly enhancing the paper's completeness.

Weakness:
1. The writing style could be further polished to improve flow.
2. While the experiments are well-designed, the soundness is slightly limited by the lack of comparison with other Reinforcement Learning (RL) algorithms beyond DPO.
3. There is also a missed part to analyze the frequency and logic behind choosing specific heads during the inference phase.

---

> ### Author Rebuttal · Authors · 2026-03-31
>
> We appreciate your thoughtful and constructive feedback of our work. Below we will respond to the raised questions.
>
>
> **Response to Weakness 1:**
>
> Thanks for the suggestion. We will carefully polish the writing to improve flow.
>
> ---
> **Response to Weakness 2 and Question 1:**
>
> Thanks for the valuable suggestion. To directly address it, we conducted an additional comparison with Multi-Objective PPO (MOPPO) following the baseline setup in [1], where the three normalized reward signals are combined with equal weights into a single scalar reward for PPO optimization. We report the results on Human Values domain:
>
>   | Method | Helpfulness | Honesty | Truthfulness |
>   |--------|:---:|:---:|:---:|
>   | Base | 0.5800 | 0.3042 | 0.1888 |
>   | MODPO | 0.6175 | 0.3477 | 0.2325 |
>   | MOPPO | **0.6764** | 0.2921 | 0.1460 |
>   | MAH-DPO Ensemble | 0.6389 | **0.3687** | **0.2478** |
>
> The results show that MOPPO achieves the highest helpfulness score, but at a substantial cost to honesty and truthfulness, leading to a less balanced multi-objective profile overall. In contrast, MAH-DPO maintains strong performance across all three dimensions simultaneously, suggesting that preserving objective-specific preference information can better support balanced multi-objective alignment than optimizing a single scalarized reward.
>
> [1] Beyond One-Preference-Fits-All Alignment: Multi-Objective Direct Preference Optimization. ACL 2024 Findings
>
> ---
> **Response to Weakness 3, Question 2, and Limitation 1:**
>
> Thanks for this important question. To clarify, our current method enables controllable inference through user-specified head weighting, rather than using a learned dynamic per-step router that automatically switches between heads during generation. Accordingly, the most relevant analysis is not head-switch frequency, but how different head-weight settings affect behavior. In Section 6.1, Table 1 compares individual heads and the ensemble head, showing that specialized heads perform strongly on their target objectives while the ensemble gives the most balanced overall profile. Figures 2 and 3 then vary the inference-time head weights and show smooth, predictable trade-offs with limited interference, rather than abrupt switches or instability. We will revise the paper to make this inference mechanism clearer and emphasize that our current control is based on user-specified head weighting, while learned or context-dependent routing over heads is a natural direction for future work.
>
> ---
> **Response to Limitation 2:**
>
> Thanks for the helpful suggestion. We introduce the multi-action-head architectural design in Section 5.1, describe the experimental setup in Section 6.1, and provide detailed training hyperparameters for each domain in Appendix B. We will consolidate them into a clearer implementation summary to make these implementation details easier to follow. We are also committed to releasing the full codebase, including training scripts, configurations, and evaluation pipelines, to support reproducibility.
>
> ---
> **Response to Limitation 3:**
>
> Thanks for the suggestion. In this paper, we focus on DPO-based alignment as a clean and efficient setting to study the multi-action-head formulation. Extending the framework to PPO-style and other RL-based optimization methods is an interesting direction for future work.

---

> > ### Author Rebuttal · Reviewer_Umiw · 2026-04-01
> >
> > Thanks to the authors, I will maintain the positive score.

---

> > > ### Author Response · Authors · 2026-04-04
> > >
> > > Thank you for confirming that our response has addressed your concerns. We sincerely appreciate your positive evaluation and your continued support of our work.

---

### Official Review · Reviewer_cJwq · 2026-03-13

**Soundness:** 3
**Presentation:** 3
**Significance:** 3
**Originality:** 2
**Overall Recommendation:** 4
**Confidence:** 4

**Summary:**

This paper proposes a unified multi‑objective alignment framework that simultaneously handles verifiable rewards (e.g., mathematical correctness) and non‑verifiable rewards (e.g., helpfulness, honesty). The framework consists of three components: 1) a standardized PRM training pipeline covering both verifiable and non‑verifiable domains; 2) Multi‑Action‑Head DPO (MAH‑DPO), which uses a shared LLM backbone with multiple objective‑specific projection heads to perform vectorized preference optimization; and 3) PRM‑guided decoding with continuing hidden states, enabling controllable test‑time alignment. The method is evaluated across three domains: mathematical reasoning, human‑value alignment, and AI tutoring.

**Compliance With Llm Reviewing Policy:**

Affirmed.

**Key Questions For Authors:**

The multi‑action head architecture appears closely related to standard multi‑task learning with a shared backbone and task‑specific heads, which is a well‑established approach. Could the authors clarify what the specific novelty of MAH‑DPO is beyond this standard formulation?

**Limitations:**

yes

**Strengths And Weaknesses:**

Strengths:
- Unified and complete framework design. The paper presents a coherent pipeline spanning PRM training, MAH‑DPO optimization, and test‑time decoding, resulting in a well‑integrated end‑to‑end alignment approach.
- Standardized PRM training across verifiable and non‑verifiable domains is a valuable practical contribution, addressing an important gap in current alignment pipelines.
- The finding of reward verifiability guides optimization strategy provides a practically meaningful insight with clear implications for how different objectives should be optimized.

Weaknesses
- Limited gains on verifiable objectives. The improvement in mathematical accuracy during training is relatively small, suggesting that the benefits of MAH‑DPO are concentrated primarily on non‑verifiable objectives.
- Limited conceptual novelty of the multi‑action head design. The multi‑action heads are implemented as replicated linear projection heads, which closely resemble standard multi‑task learning with task‑specific output heads. As a result, the architectural contribution is incremental rather than fundamentally novel.

---

> ### Author Rebuttal · Authors · 2026-03-31
>
> Thank you for your detailed and constructive feedback, each of your concerns is addressed in our response below.
>
> **Response to Weakness 1:**
>
> Thanks for the thoughtful observation. We would like to clarify that the more modest gains on verifiable objectives during training-time optimization are consistent with one of the main empirical findings of our paper, rather than a limitation unique to MAH-DPO. In Section 6.3, we show that reward verifiability affects where optimization is most effective: when rewards are precise and checkable, test-time PRM-guided decoding tends to provide larger gains, whereas noisier and more subjective objectives benefit more from training-time multi-objective alignment. Therefore, this complementarity between training-time and test-time alignment is a central design principle of our unified framework. We will revise the paper to make this point clearer.
>
> ---
> **Response to Weakness 2 and Questions:**
>
> Thanks for the question. We would like to clarify that the main contribution of our paper is the unified multi-objective alignment framework, which combines standardized PRM training, MAH-DPO, and PRM-guided decoding for controllable alignment across verifiable and non-verifiable rewards. Within this framework, MAH-DPO serves as the training-time mechanism for preserving objective-specific preference information, rather than collapsing multiple objectives into a single scalarized policy. Unlike standard multi-task learning with a shared backbone and task-specific heads, where different heads usually solve different tasks, our heads operate on the same generative task and input space, and instead separate different preference dimensions of the same behavior. This allows a single model to support objective-specific control and head mixing at inference time, while better matching the inherently multidimensional nature of human preferences. We will revise the related work section to make this distinction clearer.

---

> > ### Author Rebuttal · Reviewer_cJwq · 2026-04-04
> >
> > I thank the authors for their response to my comments. I maintain the positive score.

---

> > > ### Author Response · Authors · 2026-04-04
> > >
> > > We sincerely appreciate your feedback and the time you dedicated to evaluating our work. Thank you for your positive rating and continued support.

---

### Decision · Program_Chairs · 2026-04-30

**Decision:**

Accept (regular)

**Comment:**

I recommend Weak Accept. The reviewer consensus is clearly positive, with all reviewers giving supportive scores and little substantive disagreement. Reviewers generally agree that the paper addresses an important and practical problem in multi-objective alignment, and they view the overall framework as technically solid, well motivated, and supported by a fairly thorough empirical evaluation across multiple domains.

The main concerns were about novelty and presentation rather than soundness. In particular, reviewers noted that the multi-action-head design is somewhat incremental relative to standard shared-backbone multi-task formulations, and asked for clearer discussion of conflicting versus complementary objectives, inference-time head behavior, and broader comparisons beyond the DPO setting. The authors addressed these points well in the rebuttal through additional experiments and clarifications, and the final reviewer sentiment remained uniformly positive. Overall, this is a solid contribution with limited controversy, and I believe it merits acceptance.